# A 305-year continuous monthly rainfall series for the Island of Ireland (1711-2016)

Conor Murphy[1], Ciaran Broderick[1], Timothy P. Burt[2], Mary Curley[3], Catriona Duffy[1], Julia Hall[4],

Shaun Harrigan[5], Tom K.R. Matthews[6], Neil Macdonald[7], Gerard McCarthy[1], Mark P. McCarthy[8],

Donal Mullan[9], Simon Noone[1], Timothy J. Osborn[10], Ciara Ryan[1], John Sweeney[1], Peter W. Thorne[1],

Seamus Walsh[3], Robert L.Wilby[11]

[1] Irish Climate Analysis and Research UnitS (ICARUS), Department of Geography, Maynooth University, Ireland.

[2] Department of Geography, Durham University, Durham, DH1 3LE, UK and Department of Geographical Sciences,
University of Bristol, Bristol, BS8 2LR, UK.

[3] Climatology and Observations Division, Met Éireann, Dublin, Ireland.

[4] Institute of Hydraulic Engineering and Water Resources Management, Technische Universität Wien, Vienna, Austria.

[5] Centre for Ecology & Hydrology, Wallingford, Oxfordshire, OX10 8BB, UK.

[6] School of Natural Sciences and Psychology, Liverpool John Moores University, Liverpool, Merseyside, L3 3AF, UK.

[7] Department of Geography and Planning, School of Environmental Sciences, University of Liverpool, Liverpool, UK.

[8] Met Office, Hadley Centre, Fitzroy Road, Exeter, EX1 3PB, UK.

[9] School of Natural and Built Environment, Queen's University Belfast, N. Ireland, UK.

[10] Climate Research Unit, School of Environmental Sciences, University of East Anglia, Norwich, UK.

[11] Department of Geography, Loughborough University, UK.

*Correspondence to*: Conor Murphy (conor.murphy@mu.ie)

**Abstract**

A continuous 305-year (1711-2016) monthly rainfall series (IoI_1711) is created for the Island of Ireland. The post 1850 series draws on an existing quality assured rainfall network for Ireland, while pre-1850 values come from instrumental and documentary series compiled, but not published by the UK Met Office. The series is evaluated by comparison with independent long-term observations and reconstructions of precipitation, temperature and circulation indices from across the British-Irish Isles. Strong decadal consistency of IoI_1711 with other long-term observations is evident throughout the annual, boreal spring and autumn series. Annually, the most recent decade (2006-2015) is found to be the wettest in over 300 years. The winter series is probably too dry between the 1740s and 1780s, but strong consistency with other long-term observations strengthens confidence from 1790 onwards. The IoI_1711 has remarkably wet winters during the 1730s, concurrent with a period of strong westerly airflow, glacial advance throughout Scandinavia and near unprecedented warmth in the Central England Temperature record – all consistent with a strongly positive phase of the North Atlantic Oscillation. Unusually wet summers occurred in the 1750s, consistent with proxy (tree-ring) reconstructions of summer precipitation in the region. Our analysis shows that inter-decadal variability of precipitation is much larger than previously thought, while relationships with key modes of climate variability are time-variant. The IoI_1711series reveals statistically significant multi-centennial trends in winter (increasing) and summer (decreasing) seasonal precipitation. However, given uncertainties in the early winter record, the former finding should be regarded as tentative. The derived record, one of the longest continuous series in Europe, offers valuable insights for understanding multi-decadal and centennial rainfall variability in Ireland, and provides a firm basis for benchmarking other long-term records and reconstructions of past climate. Correlation of Irish rainfall with other parts of Europe increases the utility of the series for understanding historical climate in further regions.

**Keywords**: *Precipitation reconstruction, observations, weather diaries, Island of Ireland, variability and change.*

**1 Introduction**

Long historical weather records are essential for understanding climate variability and change, as well as contextualising extreme events, identifying emerging trends, evaluating climate models, and supporting vulnerability and risk assessments (e.g. Matthews et al., 2016). Continuous observations of precipitation in the British-Irish Isles (BI) can be traced back to 1677 – the year the first known rain gauge was developed by Richard Towneley of Burnley, Lancashire, in NW England. Since the early 1700s, at least three precipitation gauges have operated somewhere in the BI every year (Jones and Briffa, 2006). The earliest meteorological observations in Ireland began at the end of the 17[th] Century. Unfortunately, these early instrumental records, taken in Dublin by William and Samuel Molyneux, have been lost (Shields, 1983). While discontinuous records exist, systematic weather observing did not begin in Ireland until 1789 when Richard Kirwan set up a series of instruments in Dublin (Shields, 1983). Yet, with the exception of Butler et al. (1998) who analysed the record for Armagh Observatory (commencing in 1838), there has been little work on Irish precipitation measurements prior to 1850, due primarily to the lack of suitable digitised data. There have also been few assessments of qualitative descriptions from weather diaries pre-1850. Exceptions from the 18[th] Century include analyses of the diary of Thomas Neve from Derry, for the period 1711-1725 (Dixon, 1959), the diary of Isaac Butler from Dublin covering 1716-1734 (Sanderson, 2017) and Joshua Wight from Cork during June 1753 to September 1756 (Tyrell, 1995). Thus, Irish rainfall climatology over the last 300 years remains poorly understood.

The spatially variable nature of precipitation, together with changes in observer practices, gauge location and design, mean that developing reliable, long-term precipitation series can be a challenging task (Wilby et al., 2017). The assembly of regional series from individual gauges can be further hampered by changes in the network of gauges through time. Nonetheless, such regional series can provide important insights into precipitation variability and change over the course of centuries. Symons (1866) and later Nicholas and Glasspoole (1931) were among the first to construct a regional average monthly rainfall record for England and Wales extending back to 1727. Despite known homogeneity issues (due to changes in methods of rain gauge construction and siting through time), this record ultimately led to the development of the England and Wales Precipitation (EWP) series, beginning in 1766 (Wigley et al, 1984; Alexander and Jones, 2001). Similarly, Glasspoole (1925) developed a regional monthly precipitation series representing Ireland for the period 1881-1924. Tabony (1980) and Briffa (1984) developed long-term rainfall series for Irish stations, which have since been updated by others (e.g. Jones, 1983; Gregory et al., 1991; Jones and Conway, 1997). More recently, Noone et al. (2016) constructed a homogenous long-term (1850-2010) monthly rainfall network for Ireland, consisting of 25 stations together with an Island of Ireland (IoI) composite series (arithmetic mean of the 25 stations). Subsequent work used these data to evaluate severe Irish droughts from 1850 to present (Wilby et al., 2016; Noone et al., 2017), with protracted droughts being a feature of the 19[th] Century, but largely missing from recent records – particularly since the mid-1970s (Murphy et al., 2017).

An unpublished UK Meteorological Office (UKMO) note by Jenkinson et al. (1979) was recently rediscovered amongst a collection of old files following a refurbishment of Met Éireann. This note provides a continuous monthly rainfall series for IoI covering the period 1711-1977 based on documentary sources (weather diaries) and early observations. Given the painstaking work of the original authors in constructing the series, together with the possibility of extending IoI rainfall records to the early 18$^{th}$ Century, revisiting the work of Jenkinson et al. (1979) holds considerable potential for better understanding long-term Irish rainfall climatology.

Therefore, this paper (i) presents the previously unpublished Jenkinson et al. (1979) data and its constituent sources; (ii) merges the Jenkinson data with the homogenised Noone et al. (2016) series to produce a continuous monthly rainfall series for Ireland from 1711 to present (henceforth IoI_1711); (iii) assesses confidence in the early record through comparison with other long records of relevant meteorological parameters from the BI region, as well as available proxy and documentary sources; and, (iv) investigates the nature of variability and change in the reconstructed rainfall record. The rest of the paper is organised as follows. Section 2 presents the various sources of information (including the Jenkinson note) for constructing the IoI_1711 series, alongside other data used for comparative analysis. The methods for exploring variability and change are also presented. Results are described in Section 3. Section 4 provides a discussion of key findings and limitations of the new series. Finally, Section 5 closes with conclusions and some priorities for future work.

## 2. Data and Methods

### 2.1 Deriving the extended Island of Ireland (IoI_1711) series

Two data sources were used to construct the IoI_1711 series. The first is the Noone et al. (2016) IoI composite series (1850-2010) (henceforth IoI_1850 series). This dataset consists of monthly rainfall data for 25 stations across the island, each homogenised via the community standard *HOMogenisation softwarE in R* (HOME, 2013) software package and making recourse to available metadata. These homogenised data were combined by simple arithmetic mean into a composite monthly precipitation series representing the Island of Ireland (IoI). McCarthy et al. (2016) updated this composite series to February 2016, which are the data used here. The second source is the UKMO note by Jenkinson et al. (1979) (henceforth the Jenkinson series), which contains tables of monthly and annual rainfall representing IoI for the years 1711 to 1977. A copy of the original document is provided in Supplementary Information (SI).

### 2.1.1 The Jenkinson series

The Jenkinson data are presented as annual totals expressed as a percentage of the mean Annual Average Rainfall (AAR) for the period 1826-1975. Monthly values are provided as a proportion of the rainfall for each year. Annual anomalies and proportionate monthly values, presented in the Appendix of Jenkinson et al. (1979), were transcribed using double keying (by two of the authors) to minimise the risk of transcription errors. Table 1 provides an overview of the various data sources

comprising the Jenkinson series. Figure 1 maps the location of contributing sources from Ireland and the UK, while Figure 2 plots the number of constituent sources per year underpinning the series, along with annual anomalies (relative to 1826-1975) contained in the note for the years 1711-1977.

The earliest instrumental observations in the Jenkinson record originate from a rudimentary gauge operated by Thomas Neve in Derry during the period 1711-1725. Given the scarcity of observations in the early part of the Jenkinson record, at times annual totals were derived from regional series from the UK. For instance, during the period 1757-1839 data from NW England were used as a proxy Irish station. Similarly, Scottish data were used for the years 1757-1977. Prior to 1811, the number of sources is typically less than four, while for the years 1728-1756 and 1766-1791 the derived series is based on

single sources – the weather diary of Dr. John Rutty (1770), who compiled monthly and seasonal weather summaries for Dublin (1716-1765), and a regional series comprising data from NW England and SW Scotland, respectively. The latter were compiled by Jenkinson et al. (1979) using available data from the UKMO. From the 1790s onwards, instrumental observations from across IoI became available, including stations at Dublin, Kilkenny, Derry, Belfast, Armagh, Cork and Sligo.

The methods employed by Jenkinson et al. (1979) to derive annual totals and proportionate monthly values are incompletely documented, but appear similar to those of Glasspoole (1925) and Nicholas and Glasspoole (1931) in deriving regional rainfall series for the UK and Ireland (see Appendix 1 in Wigley et al. 1984 for description). For available stations, Jenkinson et al. (1979) expressed monthly values as a fraction (percentage) of the annual value for each station, and these

20 fractions were then averaged over available stations. Regional estimates of annual totals were constructed by taking an overlapping period of 20-50 years with the Glasspoole series for Ireland (1881-1924) and continuing this overlapping back to the earliest records. According to Jenkinson et al. (1979), overlapping periods were varied according to the available contributing stations with longer period records taken as optimum. For each overlapping period, estimates of AAR were derived iteratively to provide: i) station AAR; ii) station annual percentages of AAR, and; iii) regional annual percentages of

25 AAR (Jenkinson et al, 1979). Only the latter are provided in the Jenkinson note. For any station, estimates of AAR were taken as the arithmetic mean of the available annual totals. Individual years were recorded as percentages of this AAR. The mean of the percentage values for a given year across available stations was then taken as the estimate of the regional percentage of AAR for that year. Finally, Jenkinson et al. (1979) used the mean for all data years to derive the final estimate of the regional AAR (see description of methods in the original document in SI).

The diaries of Thomas Neve (Dixon, 1959), Nicholas Blundell (1968) and Dr John Rutty (1770) are critical sources for the early Jenkinson series. The Neve diary contains estimates of rainfall totals from Derry taken from a rudimentary gauge (described by Dixon (1959)). Nicholas Blundell's diary provides monthly and annual weather summaries for Crosby (Liverpool) in NW England for the years 1711-1715 and 1718-1727. The Rutty diary contains a chronology of weather in

Dublin from 1716-1765, including monthly and seasonal weather summaries. To convert the qualitative descriptions of Blundell and Rutty to a quantitative measure of rainfall, Jenkinson et al. (1979) applied a graded scaling system, similar to Brázdil et al. (2010a) and Gimmi et al. (2007), to both diaries. Scores ranging from 1 (exceptionally dry) to 9 (exceptionally wet) were used to rank individual months and aligned with seasonal summaries. Frequencies of rankings for each month

over the period 1716-1765 were derived and internal percentages for each month estimated by comparison with frequencies of monthly rankings for the period 1840-1889 (Jenkinson et al, 1979). These values were used as a percentage of AAR for each month in the period 1716-1765. Monthly and annual percentages were processed to give internal percentages for each year, with the annual percentages of AAR (1825-1977) used to represent a single station (see the original Jenkinson et al. (1979) note in SI). This combination of methods for quantitative and qualitative sources was used by Jenkinson et al. (1979)

to produce the final IoI series of annual totals, expressed as a percentage of AAR for the period 1826-1975. The final series was calibrated against Glasspoole's EWR series, although details of this step are not provided by Jenkinson et al. (1979). Moreover, Jenkinson et al. (1979) report annual anomalies relative to 1826-1975, but this AAR value does not appear in the publication.

### 2.1.2 Deriving the IoI_1711 series

The IoI_1850 series is used to estimate this missing AAR and thus to reconstruct the full series. With the exception of the 24 years prior to 1850, there is a large overlap between the normal period applied to the Jenkinson series and the IoI_1850 series. Matching annual anomalies derived from the IoI_1850 series using the normal period of 1850-1975 with the anomalies derived from the Jenkinson series for concurrent years, reveals a strong positive correlation (Spearman's r = 0.95). The time series of both sets of anomalies is plotted in Figure 2c. Resampling was used to derive estimates of long-term AAR

values from 1000 samples (with replacement) of long-term (100 year) AAR drawn from the IoI_1850 series over the period 1850-2015. Each AAR estimate was used to reconstruct annual totals and subsequently monthly rainfall totals for the years 1711-1849. Post-1850 data remain the homogenised IoI_1850 series of Noone et al. (2016). Resampling the missing AAR from IoI_1850 has two advantages. First, it ensures that the Jenkinson series is mean-adjusted to the homogenised IoI_1850 series. Second, confidence bounds can be generated for the reconstructed series to convey the uncertainty in estimating the

value of AAR. The combined dataset is named the IoI_1711 series and provides a continuous monthly rainfall record from January 1711 to February 2016. Subsequent analysis is performed on the median of the IoI_1711 series.

### 2.2 Series used for comparison with IoI_1711

Other long-term observational series of precipitation, temperature, and sea level pressure, proxy reconstructions of precipitation and modes of climate variability and circulation for the region were collated for comparison with the IoI_1711

series. Emphasis is placed on the pre-1850 period as post-1850 has already been interrogated by Noone et al. (2016). Datasets used are listed in Table 2 and the location of precipitation series are mapped in Figure 1. All reported correlations are derived using Spearman's rank correlation, with significance reported at the 0.05 level. There is a risk of circularity when

comparing with precipitation records from the UK since some of these observations and documentary sources were used directly by Jenkinson et al. (1979) to calibrate their series against EWR. Therefore, when presenting the data used, series are grouped into two categories: (I) those where a risk of circularity exists and (II) those that are deemed to be independent, based on examination of data sources.

### 2.2.1 Category I series that may not be fully independent

*England Wales Rainfall (EWR):* Nicholas and Glasspoole (1931), building on earlier work by Symons (1866), presented a regional monthly precipitation series for the UK (1727-1931). Jenkinson et al. (1979) used annual EWR data to calibrate their series for Ireland, so there are obvious circularities. However, we include the years 1727-1765 prior to the modern England and Wales Precipitation (EWP) series that begins in 1766 (see below). Early EWR data were analysed by Jones and Briffa (2006) and provided by Prof. Phil Jones. We treat the EWR series by simply appending it to the EWP series (where EWR and EWP are appended in such manner we refer to the resultant series as EWR/EWP). Wigley et al. (1984) note inhomogeneities in the annual totals in the early EWR record, whereby there is a tendency to underestimate precipitation relative to EWP prior to 1870. Moreover, Tabony (1980) suggests that prior to 1820 EWR is unreliable. Knowledge of these discrepancies is useful for assessing the IoI_1711 series. The stations comprising the early EWR series are unclear and there is a need to reconcile the early EWR series of Glasspoole (1925) with the earlier Symons (1866) work. Glasspoole (1925) mentions that from 1727-1756 at least two stations were available, and from 1757-1774 at least three. Craddock (1976) provides a comprehensive list of stations that would have been available for this period of time – certainly more than used by Glasspoole (1925). Table 3 provides a list of stations that were operational (with at least one year of data) in England (note none listed for Wales) between 1725 and 1766. This covers the early period in the IoI_1711 series up to the start of the modern EWP series when traceability is clearer. This list of stations is used to identify potential circularity with the IoI_1711 series.

*England and Wales Precipitation (EWP) Series:* The England and Wales Precipitation series from 1766 onwards (Wigley et al., 1984; Alexander and Jones, 2001) and data for key EWP stations prior to 1766 are also employed. While EWP runs from January 1766 to present, some constituent records extend into the 1700s (e.g. Carlisle, Kew, Spalding at Pode Hole). The early part of these records often contains data from the stations listed in Table 3. However, many have also undergone adjustments to correct uncovered quality issues in the early series. An overview of each series is given in the following paragraphs. The EWP series was accessed via the UKMO website (https://www.metoffice.gov.uk/hadobs/hadukp/).

*Kew Gardens Precipitation (Kew):* Routine rainfall measurements have been taken at Kew since 1871. Prior to this, annual and monthly totals were estimated by Wales-Smith (1971) back to 1697. During the years 1717-1724 and 1725-1744, the Richmond diaries and the diary of George Smith respectively, were used to derive estimates of monthly totals calibrated to observational series at Upminster, Fleet Street and Tonbridge (all in Table 3). For the remainder of the 1700s, data from

Tonbridge, Lambeth and Somerset House feature strongly and the former two are noted in the series available to EWR in Table 3. However, monthly and annual totals for 1697-1870 were later revised by Wales-Smith (1980). Data used here were provided by Todd et al. (2013).

5     *Spalding (Pode Hole) precipitation (Spald):* The early record for Spalding (1726- present) is a composite of records from the East Midlands developed by Craddock and Wales-Smith (1977) (including Southwick (Oundle) 1726-1736 and Lyndon 1737-1798 from Table 3), while much of the record for the 19$^{th}$ Century derives from South Kyme and Pode Hole thereafter. Tabony (1980) corrected the series for being overly wet during 1770-1870. The quality of the pre-1770 record is largely unknown. Data used here were provided by Todd et al. (2013).

*Carlisle Precipitation (Carl):* The Carlisle series (1757-present) for NW England was constructed as a composite of stations prior to 1872. An annual series for Carlisle was first developed by Craddock (1976) and a monthly series subsequently developed for the years 1845-1976 by Tabony (1980). The series was further extended to 1757 by Jones (1983). Todd et al. (2015) updated the Carlisle series to present, and assessed the homogeneity of the record. Data for the years 1757-1783 come

from observations taken by Dr. J. Carlyle, as listed in Table 3. Rainfall totals prior to 1790 were found to be under-representative and were increased by 24%, while annual totals during 1827-1850 were also increased by 10% (Todd et al., 2015). Data used here were provided by Todd et al. (2015).

*Oxford Precipitation (Ox):* The Radcliffe Observatory, founded in Oxford in 1772, is the longest and best documented

continuous series of temperature and rainfall at any single site in the UK. Daily observations are available from April 1814, and continuously since 1827 (Burt and Howden, 2011). A reconstructed monthly precipitation record, first developed by Craddock and Craddock (1977), exists from 1767. Prior to 1815, monthly totals are estimated from the manuscripts of Dr. Thomas Hornsby (often incomplete and fragmentary) and data from Shirburn Castle (12 miles from Oxford), together with estimates from other stations around Oxford (Stroud, Sunbury and Lambeth, the latter is noted in Table 3). The Oxford

record is unusual in that much work has been done to ensure that the entire record is homogenous, including the early measurements from the roof of the Observatory (Burt and Howden, 2011). Data used here were provided by co-author Prof. Tim Burt.

*North Atlantic Oscillation (NAO) reconstruction (L-NAO)*: Luterbacher et al. (2001) developed a monthly resolution multi-

proxy reconstruction of the NAO Index back to 1659 and seasonal reconstructions back to 1500. Full details of the reconstruction methods and constituent data sources are given by Luterbacher et al. (2001). Issues of circularity arise when using Luterbacher et al. (2001) to interrogate the IoI_1711 series because of the inclusion of Kew, Spalding (Pode Hole), EWP and other UK records in the reconstruction. During the 1700s the number of predictors falls rapidly from approximately 40 to less than 20, so these UK precipitation series are likely to be more heavily weighted in the NAO

reconstruction for this period. Data were downloaded from the Climatic Research Unit (CRU) of the University of East Anglia (https://crudata.uea.ac.uk/cru/data/paleo/naojurg/ ).

*European seasonal rainfall reconstructions (Pauling):* Pauling et al. (2006) provide a seasonal gridded (0.5$^\circ$ resolution) precipitation reconstruction for European land areas for the period 1500-1900. From 1901 to 2000 the dataset is the gridded reanalysis of Mitchell and Jones (2005). Reconstructions are developed using a variety of long instrumental precipitation series (including for Ireland and the UK; see Figure 1 of Pauling et al. (2006)) and other documentary and proxy sources. Data were downloaded from the Climatic Research Unit (CRU) (https://crudata.uea.ac.uk/cru/projects/soap/data/recon/#paul05). The earliest observations that would have been part of EWR and used by Jenkinson et al. (1979) to calibrate their series, are also used as predictors in the Pauling et al. reconstructions (e.g. Kew, Spalding). Data for grids closest to each of the 25 stations comprising the IIP network (Noone et al., 2016) were extracted and seasonal regression adjustments developed over the period 1850-2000 using each IIP station. Average seasonal scaling adjustments applied to Pauling et al. (2006) data in bridging to IIP stations were 1.03 (DJF), 0.99 (MAM), 0.91 (JJA) and 1.01 (SON). The mean of the 25 regression adjusted series were then averaged to produce an island of Ireland series from the Pauling et al. (2006) data.

*Rutty (1770) weather diary (Rutty)*: The Rutty diary provides qualitative insight to the weather conditions experienced in Ireland during the years 1716-1765. As outlined above, the Rutty diary was used extensively in the Jenkinson series. Nonetheless, drawing on the actual descriptions in the diary is useful to help contextualise conditions in the early record. A scanned copy of the Rutty diary is available online (https://archive.org/details/achronologicalh00ruttgoog).

### 2.2.2 Category II series that are independent

Nine independent datasets consisting of observational and proxy records are also employed:

*Central England Lake District (CELD) precipitation series:* Barker et al. (2004) developed a 200-year homogenous monthly composite series for the Central England Lake District (CELD) (1788- 2016). Available stations were bridged to Grasmere High Close using regression to develop seasonal adjustment factors. Much of the pre-1850 record is drawn from sites at Kendall and Keswick. Neither are key stations in the development of EWR/EWP (although Kendall was used as a station for regression development by Wigley et al. (1984)). Wilby and Barker (2016) subsequently updated the CELD series with a further revision of adjustment factors undertaken in 2017. The data used here were obtained from Wilby and Barker (2016).

*Hoofddorp (1735-1973) (Hoof):* Monthly precipitation data recorded for the Netherlands over the period 1735-1973 (Tabony, 1981; Slonosky, 2002), downloaded through the KNMI Climate Explorer website (http://climexp.knmi.nl/)

*Central England Temperature (CET) record* (Manley, G. 1974; Parker et al., 1992): Temperature data coincident with IoI_1711 is used to assess consistency of rainfall, especially in winter given that warm winters tend to be associated with wet conditions. CET data were downloaded from the UKMO website (https://www.metoffice.gov.uk/hadobs/hadcet/).

5    *Cork annual rainfall totals (Cork)*: In his list of rainfall records operating before 1780, Craddock (1976) cites a number of stations for Ireland including Derry (1711-1724), taken by Thomas Neve; Castle Dobbs, Antrim (1727), taken by A. Dobbs and annual totals for Cork (1738-1748) taken by Dr Timothy Tuckey. While the first two sources are used in the Jenkinson series, the latter for Cork is not. Annual totals for Cork (south Ireland) were published by Wakefield (1812: 207) and transcribed here. Neither the exact location nor the design of Tuckey's gauge is known.

*Westerly Index (WI):* Barriopedro et al. (2014) developed a monthly index of atmospheric circulation variability over the North Atlantic from 1685-2008. Their index is based on direct observations of wind direction from Royal Navy logbooks from 1685-1850 provided by ship movements in the English Channel. After 1850, the CLIWOC v1.5 (Garcia-Herrera et al., 2005) and the ICOADS v2.1 (Worley et al., 2005) datasets for the same area are used. The so-called Westerly Index (WI)

15    provides a measure of the persistence of westerly winds beneath the exit zone of the North Atlantic extratropical jet-stream (Wheeler et al., 2010). Before 1850, the WI contains only one record of wind direction (measured with a 32-point compass) per day, with data available for 95 percent of days (Barriopedro et al., 2014). The monthly WI used here is defined as the percentage of days per month with prevailing wind from the west (i.e. blowing from between $225^o$ and $315^o$ from true north). WI data were obtained from Prof. Dennis Wheeler.

*Paris London (PL) Index*: Cornes et al. (2013) developed the Paris London index (1692-2004) as an indicator of the state of the NAO index. This index, developed from recovered and corrected mean sea level pressure (MSLP) data from the respective cities, provides a consistent measure of westerly air flow over northwest Europe. Full details of the construction of the PL index, together with a comparison with available NAO reconstructions, are provided by Cornes et al. (2013). Only

25    seasons with no missing data in any constituent month were employed. Data were obtained from the CRU website (https://crudata.uea.ac.uk/cru/data/parislondon/).

*London SLP (L-SLP)*: Cornes et al. (2012) present a 300-year (1692-2007) daily series of MSLP for the city of London. Digitised data were transcribed from multiple sources, quality controlled, corrected and homogenised to represent daily

30    means of MSLP at standard modern-day conditions. Monthly values of MSLP are not reported when missing values exceed 20%. Here, seasonal MSLP is only derived when all three months are reported. Data were obtained from the CRU website (https://crudata.uea.ac.uk/cru/data/parislondon/).

*East Atlantic/Western Russia pattern (Eurasia 2 pattern EU2) (EU2 Index):* The EU2 index (Barnston and Livezey, 1987) measures the zonal pressure difference across central Europe and is important in describing the variability of Eurasian climate, especially during boreal winter (Luterbacher et al., 1999). The EU2 pattern is characterised by two main large scale pressure anomalies located over the Caspian Sea and Western Europe and is found to be closely related to Rossby wave propagation (Lim, 2015). Luterbacher et al. (1999) reconstructed monthly EU indices back to 1675, with the derived data for the EU2 index provided for this analysis by Prof. Luterbacher.

*Southern England tree ring precipitation reconstruction (Rinne)***:** Rinne et al. (2013) present a 400-year long (1613-2003), annually resolved May to August precipitation reconstruction for southern England developed from oxygen isotope measurements of tree ring cellulose in pedunculate oak (*Quercus robur*)**.** Using these reconstructions (1613-1893) and instrumental data (1894-2003), their derived precipitation series has been shown to be robust back to at least 1697 – the first year of the oldest existing instrumental precipitation series in England (at Kew, see above). Data were obtained from Dr. Katja Rinne-Garmston.

## 2.3 Quality assurance, variability and change

The annual and seasonal IoI_1711 series were tested for evidence of step changes in the mean and for change in variance. The Shapiro-Wilks test (Royston, 1982) confirms that the annual and seasonal series conform to a normal distribution. Given the lack of a homogenous reference series for comparison using relative methods, the Pettitt (1979) test and the Standard Normal Homogeneity Test (SNHT) for a single break (Alexandersson, 1986) were used to examine evidence for break points in the mean. Both tests were selected given their wide use in the climate literature and their ability to identify the likely year of break. Pettitt is an absolute, non-parametric test and, being rank based, is less sensitive to outliers. SNHT assumes data are normally distributed. For all tests the null hypothesis (no change point in time series) against the alternative (an upward or downward change point in a given year) was tested at the 0.05 level. It should be noted that where breaks were identified no adjustments were made to the data; instead the tests were used to identify points of interest requiring further investigation.

To test for changes in variance the annual and seasonal series were split into three 100-year sections: 1711-1810, 1811-1910, and 1911-2010. As the winter series commences in 1712 each section was incremented one year (e.g. 1712-1811 and so on). While the F-test is widely used for testing differences in variance, it is extremely sensitive to assumptions of normality (Wang et al., 2008). This is also the case for the Bartlett's test (Bartlett, 1937). Here, application of the Shapiro Wilks test to each annual and seasonal 100-year section reveals a small number of series to be non-normal. Thus, we use Levene's test which is less sensitive to departures from normality (Conover et al., 1981; Snedecor and Cochran, 1989: 252) to compare variances across each of the three 100-year sections. For all tests the null hypothesis (that each section has identical variance) against the alternative (at least one of the sections has a different variance) was tested at the 0.05 level.

Correlation between IoI_1711 rainfall and other long term series was assessed using the non-parametric Spearman's Rho test. To examine the extent of the relationship between IoI_1711 and precipitation across Europe, seasonal and annual correlations were computed between the CRU TS V4.1 gridded series (0.5 degree resolution) (Harris et al., 2013) and IoI_1711 for the years 1901/02 – 2015/16. For selected long-term series moving 30-year correlations were assessed. The 95% confidence levels for moving correlations were identified using a Monte Carlo procedure for which correlations between the observed and a set of one thousand randomly generated time series were estimated (Kokfelt and Muscheler, 2013). Confidence levels were calculated as the 2.5[th] and 97.5[th] percentiles of the moving correlation values returned by simulated series. Following Pauling et al. (2006), each time series generated by the Monte Carlo procedure has the same statistical attributes (variance, mean and lag-one autocorrelation) as the observations (Gershunov et al., 2001).

All records were standardised (rescaled to have a mean of zero and a standard deviation of one) to the period 1900-1950 to assess decadal variability and change. This period is common to all datasets and is a time of good data coverage and quality. Annual and seasonal (winter (DJF), spring (MAM), summer (JJA) and autumn (SON)) decadal moving averages were then computed for all long-term datasets. Decades are named according to their start year (i.e. 1950 denotes the ten years 1950-1959). The non-parametric Mann–Kendall (MK) test (Mann, 1945; Kendall, 1975) was used to detect monotonic trends in seasonal and annual totals. The MK test statistic ($Zs$) has mean of zero and variance of one. Positive (negative) $Zs$ indicates a positive (negative) trend in precipitation. The magnitude of $Zs$ indicates strength of the trend. Trend significance was assessed at the 0.05 level using a two-tailed test. The null hypothesis of no trend was rejected if $|Zs|>1.96$. Dependency of trends on the period of record was investigated by deriving MK $Zs$ for all possible start and end dates with a minimum of 30 years duration (e.g. Murphy et al., 2013; Hall et al., 2014).

## 3.    Results

### 3.1   The IoI_1711 series

To derive the IoI_1711 series 1000 re-samples (with replacement) of long-term (100-year) AAR were drawn from the homogenised IoI_1850 series. These range between 1039 mm to 1120 mm, with a median of 1080 mm. Each re-sample was used to estimate the annual anomalies and subsequently monthly totals from the pre-1850 Jenkinson data. Figure 3 shows the resultant IoI_1711 time-series annual and seasonal totals. The rest of our analysis is based on the IoI_1711 series calculated from the ensemble median.

Application of the Pettit test to the median annual series revealed no statistically significant step change in the mean at the 0.05 level. However, further investigation of p-values reveals a significant change at the 0.1 level (p = 0.053) in 1922. Noone et al. (2016) also found significant breaks (p < 0.1) across many of the 25 stations comprising the IoI_1850 series in the early 1920s. Consistency in the timing of these breaks across stations, along with an absence of evidence from metadata describing

widespread measurement changes across the island, suggests that this break is due to natural climate variability. The SNHT test reveals a significant step change in 1976 (p = 0.027). The timing of breaks is consistent with an increase in annual totals associated with a shift to a positive phase of the North Atlantic Oscillation around this time (Harrigan et al., 2014). Both tests also reveal a significant upward step change in winter (1864; p < 0.001) and downward step change in summer (1855; p = 0.015). Neither step change was identified by Noone et al. (2016) for the IoI_1850 series given that they both occurred so close to the start of that record.

Levene's test reveals no significant differences in variance between the three 100-year blocks when analysed for all four seasons. However, a significant difference (p = 0.040) in the variance of the three blocks is noted in the annual series. Further investigation reveals that a statistically significant difference (p = 0.016) in the variance of annual precipitation totals occurs between the periods 1711-1810 and 1811-1910, with the 1700s (134.3mm) revealing a higher standard deviation than the 1800s (109.5 mm). But neither block shows significant differences relative to 1911-2010 (standard deviation 114.6 mm).

### 3.2 Correlation of records with IoI_1711

Figure 4 shows the correlation of precipitation for IoI with other locations in Europe for the period 1901/02 to 2015/16. Non-significant correlations (p>0.05) are masked white. Significant correlations are evident across wide areas of Europe annually and for all seasons. Correlations are strongest in winter, with strong positive correlations found throughout the UK (weaker in north-western Scotland), and areas of Brittany (northern France). Strong negative correlations with IoI are also evident for winter along the western coast of Scandinavia (also present in spring and autumn). In summer, negative correlations exist between IoI_1711 and rainfall over Italy and the Balkan Peninsula.

Table 4 shows the correlation of individual long-term series with IoI_1711 annual and seasonal totals for the period 1790-2000, the longest overlap between most stations. Note that there are missing data in some series (up to 27 years for Hoofddorp, which ends in 1973). The Pauling reconstructions consistently show the strongest correlations, however, we note that these are seasonally adjusted to IoI _1850 (see Section 2.2.1). Otherwise, EWP shows the strongest correlation with IoI_1711. Annually, all long-term series show significant correlations with IoI_1711, with EWP (r = 0.67), CELD (r = 0.58) and Carlisle (r = 0.52) revealing strongest correlations (p = 0.001). L-SLP shows a modest negative correlation with IoI_1711 annual totals (r = -0.46), with weaker positive correlations evident for the PL-index (r = 0.27), the WI (r = 0.41) and L-NAO (r = 0.21). In winter, strongest correlations (all r > 0.70) with IoI_1711 are evident for EWP, Oxford and Kew. Indices representing westerly air flow show correlations with winter IoI_1711 ranging from r = 0.51 (PL Index) to r = 0.37 (L-NAO), while the EU2 index shows significant negative correlation (r = -0.67). In spring, strongest correlations are evident with EWP (r = 0.73), CELD (r = 0.64) and Carlisle (r = 0.63). The EU2 Index and L-SLP show significant negative correlation (r = -0.64 and -0.58, respectively) while the WI shows a significant positive correlation (r = 0.46). For summer,

strongest correlations are evident with EWP (r = 0.75) and L-SLP (r = -0.71). The latter indicates the regional influence of blocking on precipitation in summer. Similar, (albeit slightly weaker) correlations are evident for autumn.

Associations with selected series are further explored by examining moving 30-year correlations (Figure 5). For comparison, correlations of each variable with EWR/EWP are also included (blue line in Figure 5). In winter (Figure 5, left) significant positive correlations are evident with CET in the early 1700s and from the 1760s through to the 1940s. Both the L-NAO and WI index show time varying correlations with both IoI_1711 and EWR/EWP. Significant positive correlations with L-NAO are evident in the early and late 1700s and early 1800s, while the WI shows significant correlations from the 1760s to the 1840s, and again for the 1870s through to the 1940s. The PL-index shows more persistent significant correlation with winter precipitation. Both L-SLP and the EU2 index show persistent negative correlations with winter precipitation, however, correlations with L-SLP are weaker in the pre-1850 record. There is strong coherence of correlations of selected series with both IoI_1711 and EWR/EWP, with the exception of the early EWR series when correlated against L-SLP and the EU2 index.

For summer (Figure 5, right), moving 30-year correlations are typically more variable than for winter. CET shows negative correlations with the precipitation series, with periods of significant correlations intermittent throughout the record. Stronger correlations are evident with EWR/EWP than for IoI_1711. Moving 30-year correlations with L-NAO tend to be weak and non-significant in summer, with the exception of the latter half of the 18[th] Century (significant positive correlation). WI shows significant positive correlations up to the 1720s and from the 1750s to the end of the record. There is an increasing tendency in the value of the correlation coefficients of both IoI_1711 and EWR/EWP with the WI. Persistently strong and significant negative correlations are apparent for L-SLP throughout the record, indicating the importance of blocking (high SLP) in the regional summer precipitation regime. The EU2 index also shows significant negative correlations throughout the record for IoI_1711.

### 3.3 Comparison of decadal variability

The following paragraphs examine the consistency of annual and seasonal decadal moving averages for IoI_1711 with other long-term series from Table 2. Table 5 distils the key findings for each season.

### 3.3.1 Annual

Figure 6a compares decadal moving averages for each long-term precipitation series. Strong agreement between all series is evident. In the pre-1766 record, EWR reveals the lowest decadal mean annual totals, consistent with concerns that EWR underestimates early totals (Jones and Briffa, 2006). In the IoI_1711 annual series the most recent decade (2006-2015) is the wettest. Indeed, the last 50 years has seen the most persistent period of decadal mean annual totals above the 1900-1950

mean. This is also the case for the Pauling data. The driest decade in the IoI_1711 annual series was 1740-1749. The 1740s was also notably dry in the other long series covering this period. For instance, the 1740s was the driest decade on record for Spalding and second driest for Kew (after 1893-1902). Although the 1740s were notably dry in the Hoofddorp series, decades commencing from the 1790s through to the early 1800s were even drier. Figure 6b plots the early annual totals from Cork taken by Dr T. Tuckey relative to the raw IoI_1711 annual totals. These data confirm that the 1740s were notably dry. In both series, 1740 was the driest year of the decade, followed by 1747. Indeed, in the entire IoI_1711 series, 1740 is the second driest year after 1788.

### 3.3.2 Winter (DJF)

Winters (DJF) are named by the year in which January falls, so that winter 1712, for example, comprises December 1711 together with January and February of 1712. Figure 7 plots decadal mean totals for all winter series standardised to the period 1900-1950. Consistency of all precipitation series (Figure 7a) from about 1790 onwards is noted. Prior to this there is divergence between the available precipitation series. Evident also is the long-term increasing trend in winter decadal mean precipitation totals across all series. Despite this increasing trend, the decade 1730-1739 was the wettest in the IoI_1711 winter record. Other long-term UK precipitation series that extend back this far, together with the Pauling series, do not show a remarkably wet decade. However, the 1730s are well known for their exceptional warmth within the CET record (Jones and Briffa, 2006), which we also note shows strong coherence with IoI_1711 throughout the record (Figure 7b). Additionally, the highest decadal value in the WI winter series is found for the 1730s and the winter L-NAO is in a positive phase at this time too.

Table 6 provides the seasonal summaries of Rutty (1770) during the 1730s. Evident for winter is the description of each year as open (frost free) and warm. In 1733 reference is made to primroses and violets blooming at Christmas. The only frosty winter is noted in 1739. While winter 1730 is noted as dry, each subsequent winter in the decade (with the exception of 1737 and 1739) is described as wet, wet and windy, or wet and stormy, which is consistent with strong westerly flow. While a number of independent series support the view that winters in the 1730s were remarkably wet in the IoI_1711 series, confidence is low in the winter decadal means from 1740 to 1785. Within the context of the entire series decadal means for 1740 to 1785 are exceptionally low with little evidence for such dry winter conditions in other long precipitation series. We note that decadal means in the CET record are cold during this time, which is consistent with a negative winter NAO and WI. While dry conditions would be expected under such winter dynamics, the deviation between CET and IoI_1711 decadal means is largest during this period. At this time, the record of Jenkinson et al. (1979) is dependent almost entirely on qualitative information from weather diaries. Despite being biased too dry, it is likely that the decade 1777-1786 was at least among the driest decades in the winter series. This is consistent with the driest winter decade in EWP (1776-1785) and Carlisle (1776-1785). The years 1776-1785 rank amongst the top 15 driest decades in the Pauling series, while closely similar years (1781-1790) are also the driest winter decade in the Oxford record. The EU2 index is also markedly positive at

this time. From 1790 onwards confidence in the IoI_1711 winter record is strengthened by the high degree of consistency with all other series.

### 3.3.3 Spring (MAM)

The pre-1850 IoI_1711 record shows substantial decadal variability (Figure 8). In IoI_1711 the wettest spring decade occurs at the beginning of the series (1715-1724), while the driest is 1831-1840. The latter is consistent with EWP (driest spring decade 1833-1842) and with Oxford (third driest decade 1831-1840). Figure 8a shows strong agreement in decadal totals throughout the record from 1740 across all precipitation series. However, the Pauling series does not show the same variability as in observed series throughout the 1700s. Prior to the 1740s there are few sources for comparison. Nonetheless, there is good coherence between Spalding, EWR and IoI_1711 during this period. Strong agreement is also evident between the WI, the EU2 index and IoI_1711. A positive WI index coincides with wet spring conditions during the earliest decades of the 1700s, at the turn of the 20[th] Century (1895-1910), during the 1950s and 1960s, and the last quarter of the 20[th] Century (from the mid-1970s). Conversely, predominance of negative NAO conditions during 1715-1724 is inconsistent with reconstructed exceptionally wet spring conditions (although we note that negative NAO conditions can result in diverse climate patterns). Neither the PL index nor L-SLP series contain sufficient data for the period to place much confidence in the comparator series. Throughout the latter half of the 1700s spring decadal means are almost always below the 1900-1950 mean, consistent with persistent negative NAO conditions and negative WI for the period.

### 3.3.4 Summer (JJA)

The main feature of the IoI_1711 pre-1850 record is the persistence of wet summers throughout much of the latter half of the 18[th] Century (Figure 9). The 1750s (1750-1759) is the wettest IoI_1711 decade. Decadal mean summer totals for this period are also notably wet in other long-term precipitation series. For instance, 1767-1776 is wettest in the Kew series (although it should be noted that the period is not as wet and is less protracted at Kew relative to other long-term series). The years 1755-1764 mark the wettest summer decade in the Hoofddorp series. The mid-1750s were also exceptionally wet at Spalding although the 1870s were marginally wetter. We note that the Pauling series does not show wet conditions throughout this period.

Figures 9b and 9c plot IoI_1711 summer decadal means for L-SLP alongside indicators of westerly flow (L-NAO, PL index and WI). The wet 1750s coincides with record low MSLP at London. Much of the latter half of the 18[th] Century is associated with strong westerly airflow, with a persistently positive summer NAO and markedly positive values for the PL index. The EU2 index is also negative throughout the latter half of the 18[th] century. Unfortunately, the summer WI has a lot of missing data for this period. Table 7 shows the seasonal summaries provided by Rutty (1770) for the 1750s. With the exception of 1759 all are described as wet. Figure 9d compares IoI_1711 with Kew, Spalding and the Rinne et al. (2013) tree ring reconstructions for southern England for the months May through August. The tree ring reconstructions are consistent with

Kew in terms of the wetness of the period, although later decades are wetter (e.g. decades commencing in 1836 and 1877), suggesting that southern England, while wet throughout the latter half of the 18th Century, was not as wet as other parts of the BI. The driest summer decade of the IoI_1711 series is recorded for 1968-1977.

### 3.3.5 Autumn (SON)

Autumn decadal means show strong coherence across all long-term precipitation series (Figure 10). Of note is the large decadal variability in autumn series prior to 1900, with high decadal mean totals prominent during 1750-1870. The wettest decade in the IoI_1711 autumn series is 1770-1779, which partly overlaps with the wettest at Carlisle (1766-1775), Oxford (1767-1776) and Spalding (1768-1777). The 1770s are also notably wet in the Pauling series. The driest autumn decade in the IoI_1711 series is 1745-1754, which matches closely with Kew (1745-1754), Spalding (1748-1757) and the early EWR series (1748-1757). In autumn, there is a strong negative correlation with L-SLP (r = -0.65), with high mean L-SLP in the years 1745-1754 coinciding with dry rainfall conditions across many stations, and low L-SLP values in the 1770s consistent with wet autumn conditions at the time. Furthermore, the driest autumn decade (1745-1754) coincides with anomalously high values for EU2 index at this time, while the wettest autumn decade (1770s) coincides with a markedly low EU2 index, adding confidence to this finding.

### 3.4 Monotonic trends in the IoI_1711 series

Trends in annual and seasonal totals for the IoI_1711 series were assessed for all possible start and end dates with a minimum record length of 30 years (Figure 11). The annual series shows a significant increasing trend, but only for records ending after 2000. The largest MK Zs values in the annual series are found for tests commencing before 1850. In winter, exceptionally large values are apparent for tests commencing prior to 1790. Significant increasing trends are also found for series commencing between 1790 and 1850. However, trends commencing post-1850 are insignificant, with weak increasing and decreasing trends apparent. Trends in spring are marked by variability, with a lack of persistence in either trend magnitude or direction. Largest MK Zs values are found for tests commencing in the 1760s and 1770s. Series beginning after 1850 show weak increasing and decreasing trends. In summer, significant decreasing trends are found for series commencing before 1900. The most notable decreasing trends are found for series starting in the mid-1700s, consistent with wet summers during this period. It is evident that outside of the 1700s decreasing trends have only emerged as significant since the 1970s. For trends commencing post 1950 (the period of record with most digital data), significant increasing trends are found. These are likely an artefact of the short record and highlight that even 60 years of data can yield trends unrepresentative of longer records. Finally, autumn is marked by weak and variable trends. Significant increasing trends are evident for tests commencing in the late 19th to early 20th Century, but do not persist.

## 4. Discussion

This paper develops a 305-year continuous rainfall series for the Island of Ireland that extends back to the late Maunder Minimum. The series was derived by merging previously unpublished work by Jenkinson et al. (1979) with the more recently derived and quality assured IoI rainfall of Noone et al. (2016). Given the variety of sources used (both observations and documentary sources), the changing number of contributing sources and stations, changing measurement apparatus and techniques, together with the difficulty in penetrating the full details of methods used by Jenkinson et al. (1979), the pre-1850 record must be treated with caution. For example, rain gauges in use during the 1700s and early 1800s were experimental and largely unrecognisable by modern standards. The gauge used by Thomas Neve in Derry is described as being 12 inches in diameter and fixed to the ridge of his house, draining through a funnel and tin pipe into a loft and collected by a large glass bottle (Dixon, 1959). Clearly, such gauges would be prone to considerable under catch. Indeed, it was not until the pioneering work of G.J. Symons, documented in the British Rainfall series from 1860 (Walker, 2010), that more widespread use of standardised gauges began (Pedgley, 2002). Additionally, it is unclear whether Jenkinson et al. (1979) made allowances for the adoption of the Gregorian calendar, which resulted in an advance of 11 days in September 1752. Jones and Briffa (2006) state, in relation to EWR, that the lack of any major seasonal cycle in precipitation over the region minimises concern about the calendar adjustment. Moreover, much of the IoI_1711 series during the 1700s is derived from weather diaries. While translating qualitative descriptions of weather into quantitative estimates of rainfall is useful for exploring variability, the actual rainfall totals must be treated as highly uncertain. Nonetheless, comparison with other long records helps to build confidence in the early record and identify aspects that warrant further investigation. Remarkably, for both spring and autumn, decadal mean totals from IoI_1711 show strong coherence with other long-term series throughout the full period of record.

Confidence is low in the winter IoI_1711 series prior to 1790 when records are likely too dry for the period 1740 to 1785. Much of this part of the record is informed by qualitative descriptions from weather diaries or by data from the UK used to represent Irish rainfall. While confidence in actual rainfall totals is low, it is likely that much of this period was indeed very dry. Moreno-Chamarro et al. (2017) highlight that exceptional wintertime conditions during the Little Ice Age arose from sea ice expansion and reduced ocean heat losses in the Nordic and Barents seas, driven by a multi-centennial reduction in the northward heat transport by the subpolar gyre (SPG). Anomalous easterlies over Western Europe deflected the westward flow of warm, moist air masses away from the continent, while increasing the frequency of wintertime blocking events. Such conditions are associated with persistent cold spells and potentially large snow accumulations (Moreno-Chamarro et al., 2017). Jones and Briffa (2006) highlight the exceptional cold and dryness of the early 1740s for the British-Irish Isles, with the impacts on Irish society and 'The Forgotten Famine' documented by Dickson (1997) and Engler et al. (2013). The 1740s are remarkably dry in the IoI_1711 winter and annual series, while 1740 stands out as exceptionally dry in the early annual totals from Cork taken by Dr. Timothy Tucker.

The weather diary of Joshua Wight from Cork offers further insight into conditions of the 1750s. In an analysis of that diary, Tyrell (1995) notes the severe cold of winters during the years 1753-1756 and, in particular, the notably low frequency of wet days during the winter of 1754-1755. A higher frequency of northerly and easterly winds is cited for the winter half year (October to March) of this period, which is typically associated with fewer wet days and longer dry spells. Tyrell (1995) also notes the middle and high latitudes were strongly affected by volcanic activity during the 1750s, with Lamb's dust veil index for these latitudes peaking between 1753 and 1756, following the eruption of Katla in 1755-1756 (Tyrell, 1995).

While the 1740s are the driest decade in the IoI_1711 annual series, the years 1777-1786 mark the driest winter decade. Similar years are also driest in several long-term precipitation records in the UK (e.g. EWP (1776-1785), Carlisle (1776-1785), Oxford (1781-1790)). Küttel et al. (2011) examined SLP reconstructions back to 1750 and found that the 1780s (in addition to the 1750s and 1760s) were marked by a high frequency of cold and dry circulation conditions during winter. Additionally, this period is marked by relatively low values of the Westerly Index, strongly negative phase of the NAO Index and exceptional cold in the CET record – all consistent with low rainfall. The period is also coincident with the Laki eruption of 1783-1784, with winters following the eruption being amongst the most severe on record in Europe and North America (Thordarson and Self, 2003). Rather than volcanic forcing, some have argued that the exceptional conditions of these years was due to natural variability related primarily to a combined negative phase of the NAO and an El Niño-Southern Oscillation (ENSO) warm event – similar to conditions during the cold and snowy European winter of 2009-10 (D'Arrigo et al., 2011). Others assert that a combination of volcanic forcing and natural variability may have played a role (Schmidt et al., 2012). Winter 1784-85 and 1785-86 both rank among the ten driest winters in the entire IoI_1711 record and are also notable in the EWP series (1784-85 ranks in the top ten driest winters, with 1785-86 ranked as 13[th] driest for the period 1766-2016). Despite being exceptionally dry, winter 1783-84 also saw significant flooding across Europe and in Ireland, particularly in January following heavy snowfall (Brázdil et al., 2010b).

Within the IoI_1711 winter series the 1730s are identified as the wettest decade in the entire 305-year series. Rather than an individual year in the 1730s standing out as remarkable, the decade is notable for persistently wet conditions. Confidence in the 1730s being exceptionally wet is strengthened by concurrent and almost unprecedented warmth in the CET (Jones and Briffa, 2006), glacial advance throughout Scandinavia (Nesje et al., 2008), and notably enhanced westerly air flow (Barriopedro et al. 2014), which are all consistent with wintertime NAO-type forcing. After 1790, when early Irish observations become available, IoI_1711 shows improved consistency with other long-term observational and proxy records. Indeed, when the Pettit and SNHT tests are applied to the winter series commencing in 1790, the step change previously identified in 1864 is no longer evident, pointing to the increase in available data and the overly dry nature of the pre-1790 record as the cause of the identified break. Rather, an upward step change is identified in 1909 (p-value = 0.002), with the same year associated with an abrupt increase in EWP (1766-2016) (p-value < 0.001).  Consistency across both series,

together with standard instrumentation and methods of observation by this time suggests that this change point is due to regional climate variability.

While summer decadal mean totals show strong coherence with other long-term records, a break in the mean is revealed in 1855. This break point towards the middle of the series likely reflects real climate non-stationarity; with exceptionally wet summers of the mid to late 1700s and very dry summers of the 1970s near the end of the record. However, it is also possible that the break is associated with increased data availability as George J. Symons formalised an early rainfall network for the island. The exceptional summer wetness of the IoI_1711 series in the mid-1700s is supported by oxygen isotope tree-ring reconstructions for southern England (Rinne et al., 2013). The period is also noted as exceptionally wet in tree-ring reconstructions (based on tree-ring widths) by Wilson et al. (2013) who report the mid-1700s as among the five wettest 20-year periods since 950 AD for the months March-July in southern-central England. The Old World Drought Atlas (Cook et al., 2015) also identifies the mid-1700s as notably wet in the context of the last 1000 years. Relative to other long-term precipitation observations (e.g. at Kew), this wet period seems to be greater in magnitude and duration in the IoI_1711 series. In his analysis of Joshua Wight's diary, Tyrell (1995) highlights that summers of the mid-1750s were wetter than recent years, noting in particular that the number of wet days reported for summer 1756 were significantly higher than the average for the modern regime. Multiple lines of evidence thus add confidence to the very wet summers of this period in the IoI_1711 series. In passing, we also note that summer 1816, the infamous 'year without a summer' following the Tambora eruption of 1815 (Luterbacher and Pfister, 2015), does not rank as notably wet in the IoI_1711 series (rank 53[rd] wettest). In line with Veale and Endfield (2016) we find that summer 1817 was wetter, ranking 4[th] wettest in our series.

The IoI_1711 series considerably extends our understanding of the rainfall regime of Ireland. Monotonic trends derived for the IoI_1711 series reveal large variability in both magnitude and direction, depending on the period of record assessed. Winter records commencing before the 1850s show statistically significant ($p<0.05$) increasing trends. Tests commencing between 1850 and 1900 show statistically insignificant positive trends. Non-significant negative trends are evident for tests commencing after 1900. It is thus evident that the statistically significant increasing trend in winter rainfall is due to cold and dry conditions in the pre-1850 record. This is also the case for EWP (not shown). Casty et al. (2007) also find that European winter precipitation prior to 1850 is lower than during the twentieth century. The early records in both IoI_1711 and EWP draw upon descriptions from weather diaries and early and experimental rain gauges that may be prone to under-catch, particularly during cold, snowy conditions. It is, therefore, possible that biases in measurement during cold conditions affect the magnitude and significance of trends in winter rainfall in long-term rainfall series. We intend to explore this potential temperature-dependent bias and influence on winter rainfall trends using ocean atmosphere models in a future study. Furthermore, Küttel et al. (2011) highlight that in winter only a small part of the observed changes in rainfall across Europe over the past 250 years are due to changes in frequencies of circulation types. Rather, within type changes (variations in the relationship between patterns of large scale circulation and associated climate) are dominant (see Küttel et al., 2011).

Summer rainfall reveals statistically significant (0.05 level) decreasing trends, but only for tests commencing before the 1900s. Trends derived from records covering the period of digitised data (i.e. 1940s onwards) are unrepresentative of the long-term, thus illustrating; i) the importance of multi-centennial records (Burt et al., 2016), and; ii) that monotonic linear trend models are inadequate descriptors of IoI rainfall behaviour.

Finally, a novel aspect of this work has been the use of multiple long-term observational and proxy sources to help assess confidence in the early IoI_1711 series. A key challenge was identifying where circularity exists in comparing with other sources. This was compounded by a lack of transparency about *which* sources were used in early records – particularly the EWR series used to calibrate the Jenkinson et al. (1979) data, and indeed the lack of detail on *how* exactly the Jenkinson
series was calibrated. Such issues highlight the importance of carefully documenting data sources in the development of regional series. Unfortunately, this was not always the case for early efforts at developing regional rainfall series. Independent, quality-assured series such as the CET, WI, L-SLP and the PL-index thus provide invaluable datasets for building confidence in long precipitation records in the region. In addition to the above independent sources, the gridded SLP dataset derived by Küttel et al. (2010) is another independent source which we did not employ here, but could prove
useful in future work. Finally, work is ongoing by Met Éireann and the Met Office National Library and Archive to digitally scan and improve the accessibility to a wide range of historical meteorological publications. For UKMO these are available through an open archive at https://digital.nmla.metoffice.gov.uk/. This resource may prove of considerable value to future climatological studies, not least in helping fill some of the gaps in our understanding of the original data and methodology for the Jenkinson Ireland series.

**5.   Conclusion**

The IoI_1711 series yields valuable insights into the long-term rainfall regime of Ireland from the late Maunder Minimum to present. In particular, the series offers an opportunity to further investigate the effects of multiple forcings (volcanic, solar, greenhouse gases, natural variability) on the rainfall climatology of a sentinel location in Europe. Our analysis shows that decadal variability may be substantially larger than previously thought from digital records. For spring, summer and autumn,
strong coherence with other long-term observational and proxy series increases confidence in the derived record. The most recent decade (2006-2015) is identified as the wettest in the 300-year annual series. While confidence in the winter series is low prior to 1790, the early record presents compelling evidence of exceptionally dry (1777-1786) and wet (1730s) winters. The long record reveals statistically significant increasing trends in winter rainfall and statistically significant decreasing trends in summer rainfall. However, we caution that the former may be influenced by temperature-related biases in the early
record leading to under-catch of snowfall. In all seasons, trends derived from the period with the most widely available digital records (i.e. 1940s onwards) are not representative of long-term trends. The continuous 305-year series developed here has many potential uses. These include a more comprehensive description of multi-decadal and multi-centennial rainfall

variability in Ireland, contextualising contemporary weather extremes, and inferring the underlying drivers of climate variability and change in the region.

Opportunities for future work to address uncertainties in the pre-1850 record include:

- Statistical and dynamical reconstruction of winter rainfall in the IoI_1711 and other long-term regional series using available independent observational sources and dynamical climate models. The Westerly Index, London Sea Level Pressure and the Central England Temperature Record could support these efforts given their strong correlation with regional winter rainfall.

- Unfortunately, oxygen isotope-based tree ring reconstructions for IoI are not yet available. Here we used reconstructions from southern England. A pilot study by Vallack et al. (2016) confirms the potential of developing a composite summer precipitation reconstruction from oak tree cores (*Quercus robur* and *Quercus petraea* L.) in Ireland, while Galvin et al. (2014) highlight the dendrochronological potential of Taxux Baccata (Yew) in southwest Ireland. It is important that this work is pursued to interrogate the exceptionally wet summers of the mid-1700s identified here.

- Ireland has a rich history of weather observing, but much of the early data remain as paper records. Future work should prioritise the digitisation and rescue of these data (e.g. Ryan et al. 2018) to maximise the utility of early observational records. There are also additional weather diaries held by Met Éireann that could help shed further light on the rainfall of the 1700s, while transcription of additional navy log books from Cork and Dublin would also be of high value.

**Supplementary Information**: The original Jenkinson et al. Met Office Note as .pdf © Crown Copyright 1979. Information provided by the National Meteorological Library and Archive – Met Office, UK.

**Data Availability**: The IoI_1711 series can be downloaded from Met Éireann's website at http://www.met.ie/cms/assets/uploads/2018/01/Long-Term-IIP-1711-2016.zip. The transcribed annual series from Cork (Wakefield, 1812) are available from the corresponding author.

### Acknowledgements

The authors are indebted to Professor Arthur Jenkinson and colleagues for their original work. An obituary marking Jenkinson's important contributions to meteorology and climatology is available from Lawson (2005). We are also grateful to Met Éireann colleagues for bringing the Jenkinson (1979) paper to light. We thank Prof. Phil Jones for providing the EWR series, Prof. Dennis Wheeler for providing the Westerly Index and to Dr. Katja Rinne-Garmston for providing her tree precipitation reconstruction. We also thank Dr. Linden Ashcroft and Prof. Jürg Luterbacher for their constructive reviews

which helped improve our manuscript. We also thank Prof. Luterbacher for providing the EU2 Index. C.M. was funded by the Irish Environmental Protection Agency under grant no. 2014-CCRP-MS.16.

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

| Years | Source | Description of the data | Reference |
|---|---|---|---|
| 1711-1725 | Derry, NW Ireland | Monthly and annual totals extracted from the weather diary of Thomas Neve. | Dixon (1959) |
| 1711-1715 | Crosby, NW England | Monthly and annual totals extracted from Nicholas Blundell's Diurnal.[1] | Blundell (1968) |
| 1716-1717 | Crosby, NW England | Monthly and annual totals extracted from Nicholas Blundell's Diurnal.[1] | Blundell (1968) |
| 1718-1727 | London, SE England | Monthly and annual totals extracted from Nicholas Blundell's Diurnal.[1] | Blundell (1968) |
| 1726-1727 | Antrim, NE Ireland | Monthly and annual totals. | Met Office |
| 1716-1765 | Dublin, E Ireland | Brief monthly and seasonal summaries extracted from a chronological history of the weather in Dublin 1716-65.[2] | Rutty (1770) |
| 1757-1839 | NW England | Monthly and annual totals. Regional series used as a single Ireland station. | Met Office |
| 1792-1839 | SW Scotland | Monthly and annual totals. Regional series used as a single Ireland station. | Met Office |
| 1757-1977 | Scotland | Monthly and annual totals. For completeness data was combined with the Glasspoole series from 1868 onwards. | Met Office/Glasspoole (1925) |
| 1792-1839 | Dublin, E Ireland | Monthly and annual totals. Supplemented by MS data available from the Met Office. | Dixon (1953) |
| 1813-1830 | Kilkenny, SE Ireland | Monthly and annual totals. | Met Office |
| 1823-1824 | Dublin, E Ireland | Monthly and annual totals. | Met Office |
| 1795-1801 | Derry, NW Ireland | Monthly and annual totals. | Met Office |
| 1814-1815 | Belfast, NE Ireland | Monthly and annual totals. | Met Office |
| 1818-1977 | Belfast, NE Ireland | Monthly and annual totals. | Met Office |
| 1836-1977 | Armagh, NE Ireland | Monthly and annual totals. | Met Office |
| 1825-1832 | Cork, S Ireland | Monthly and annual totals. | Met Office |
| 1836-1977 | Cork, S Ireland | Monthly and annual totals. | Met Office |
| 1833-1977 | Sligo, NW Ireland | Monthly and annual totals. | Met Office |
| 1840-1977 | Ireland | Monthly and annual totals. Combined with the Glasspoole series from 1881 onwards. | Met Office/Glasspoole (1925) |
| 1940-1945 | United Kingdom | Annual rainfall maps (with percentage isopleths) | Met Office |
| 1946-1948 | Ireland | Annual rainfall maps (with percentage isopleths) | Irish Meteorological Service |
| 1940-1948 | Ireland | Monthly and annual totals recorded at ten long-term stations throughout Ireland (listed below*). | Tabony (1980) |
| 1949-1977 | Ireland | Monthly and annual totals recorded at various stations throughout Ireland. | Irish Meteorological Service |

**Table 1 Description of the data used to produce the Jenkinson series of monthly and annual rainfall, 1711-1977. Derived values were subsequently converted to internal percentages of the annual rainfall for each station. For both [1] and [2] this was achieved using a ranking method to classify rainfall amounts (e.g. wet/dry). The frequency of the ranked events for each month was recorded. Using a comparable period of reference, internal percentages (per ‰) were assigned to the rankings to determine internal percentages for each year. *Markree, Valentia, Shannon, Birr, Dublin, Cork, Waterford, Armagh, Londonderry, Belfast**

| Series Name (Abbreviation used) | Years | References |
|---|---|---|
| England and Wales Precipitation (EWP)* | 1766-2016 | Wigley et al. (1984); Alexander and Jones (2001) |
| England and Wales Rainfall (EWR)* | 1725-1765 | Nicholas and Glasspoole (1931); Jones and Briffa (2006) |
| Central England Lake District Precipitation (CELD)* | 1788- 2016 | Barker et al. (2004); Wilby, 2016; revised version CELD 20170203 (Wilby, pers comm.) |
| Oxford Precipitation (Ox)* | 1766-2016 | Craddock and Craddock (1977); Burt and Howden (2011) |
| Carlisle Precipitation (Carl)* | 1757-2001 | Craddock (1976); Jones (1983); Todd et al. (2015) |
| Kew Gardens Precipitation (Kew)* | 1697-2016 | Wales-Smith (1971); Wales-Smith (1980) |
| Spalding Precipitation (Spald)* | 1726-2016 | Craddock and Wales-Smith (1977); Tabony (1980) |
| Hoofddorp precipitation series (Hoof) | 1735-1973 | Tabony (1980) |
| Pauling et al. (2006) seasonal precipitation reconstructions (Pauling) * | 1500-2000 | Pauling et al. (2006) |
| Tree ring reconstruction of southern Britain summer (MJJA) precipitation (Rinne) | 1613-2003 | Rinne et al. (2013) |
| Central England Temperature Record (CET) | 1659-2016 | Manley (1974); Parker et al., (1992) |
| Monthly NAO reconstruction (L-NAO)* | 1659-2000 | Luterbacher et al. (2001) |
| Paris-London Westerly Index (PL index) | 1692-2004 | Cornes et al. (2013) |
| London Sea Level Pressure (L-SLP) | 1692- 2007 | Cornes et al. (2012) |
| East Atlantic/Western Russia pattern (EU2 Index) | 1675-1995 | Luterbacher et al. (1999) |
| Westerly Index (WI) | 1685-2008 | Wheeler et al. (2010); Barriopedro et al. (2014) |
| Cork Annual Totals (Cork) measured by Dr. T. Tuckey | 1738-1748 | Wakefield (1812) |
| A chronological history of the weather in Dublin 1716-1765 and associated pressure measurements for Dublin* | 1716-1765 | Rutty (1770); Dixon (1969) |

**Table 2 List of data sources used for comparison with the reconstructed IoI_1711 series. Given are the full data set name and its abbreviation as used in the text and subsequent figures, the years for which data are available and the primary references for each dataset. An asterisk identifies series that are likely to be have a risk of circularity.**

| Region | Station | Observer | Years |
|---|---|---|---|
| Devon/Cornwall | Plymouth | Dr. J. Huxham | 1725-1752 |
| Devon/Cornwall | Penzance | Rev. W. Borlase | 1759-1772 |
| London | Crane Court | Mr F. Hauksbee | 1725-1733 & 1744-1781 |
| London | Tonbridge | Mr. J. Hooker | 1734-1764 |
| London | Lambeth | MS. Of G.J. Symons | 1765-1769 |
| London | Gravesend | Mr. J Hooker | 1728-1733 |
| East Anglia | Norwich | Mr. W. Arderon | 1750-1762 |
| East Midlands | Southwick (Oundle) | Mr. G. Lynn | 1726-1741 |
| East Midlands | Lyndon (Rutland) | Mr. T. Barker | 1737-1798 |
| Yorkshire | Halifax | Dr. Nettleton | 1725-1727 |
| Yorkshire | Darlington | Mr. Forth | 1734-1742 |
| Yorkshire | Pickering | T. Robinson | 1736-1749 |
| Yorkshire | Malton | NA | 1736-1741 |
| Carlisle | Carlisle | Dr. J Carlyle | 1757-1783 |
| Notts and Derby | Chatsworth | Lord G. Cavendish | 1761-1813 |
| Liverpool Area | Hulme, Manchester | Mr. G. Lloyd | 1765-1769 |

**Table 3 List of rainfall stations recording in the UK for the years prior to the commencement of the contemporary EWP series (i.e. pre 1766) which would have been available to Nicholas and Glasspoole (1931) in constructing the EWR series. Stations are taken from those listed in Craddock (1976).**

|  | Winter (DJF) | Spring (MAM) | Summer (JJA) | Autumn (SON) | Annual |
|---|---|---|---|---|---|
| EWP | **0.85\*\*** | **0.73\*\*** | **0.75\*\*** | **0.68\*\*** | **0.67\*\*** |
| Kew | **0.70\*\*** | **0.45\*\*** | **0.52\*\*** | **0.51\*\*** | **0.43\*\*** |
| Carl | **0.64\*\*** | **0.63\*\*** | **0.65\*\*** | **0.60\*\*** | **0.52\*\*** |
| CELD | **0.67\*\*** | **0.64\*\*** | **0.65\*\*** | **0.64\*\*** | **0.58\*\*** |
| Spald | **0.56\*\*** | **0.43\*\*** | **0.57\*\*** | **0.48\*\*** | **0.44\*\*** |
| Ox | **0.72\*\*** | **0.54\*\*** | **0.53\*\*** | **0.50\*\*** | **0.50\*\*** |
| Hoof | **0.46\*\*** | **0.41\*\*** | **0.42\*\*** | **0.31\*\*** | **0.29\*\*** |
| CET | **0.45\*\*** | 0.00 | **-0.34\*\*** | **0.24\*\*** | **0.20** |
| Pauling | **0.96\*\*** | **0.95\*\*** | **0.94\*\*** | **0.93\*\*** | **0.76\*\*** |
| L-SLP | **-0.58\*\*** | **-0.58\*\*** | **-0.71\*\*** | **-0.63\*\*** | **-0.46\*\*** |
| PL Index | **0.51\*\*** | **0.38\*\*** | **0.32\*\*** | **0.39\*\*** | **0.27\*\*** |
| WI | **0.45\*\*** | **0.46\*\*** | **0.47\*\*** | **0.40\*\*** | **0.41\*\*** |
| L-NAO | **0.37\*\*** | 0.07 | **0.14\*** | **0.43\*\*** | **0.21\*\*** |
| EU2 Index | **-0.67\*\*** | **-0.64\*\*** | **-0.60\*\*** | **-0.57\*\*** | **-0.50\*\*** |
| Rinne |  |  | **0.51\*\*** |  |  |

**Table 4 Spearman's rank correlation of IoI_1711 seasonal and annual totals with other long-term records for the period 1790-2000. Correlations significant at the 0.05 level are shown in bold, \* indicates significant correlations at the 0.01 level and \*\* at the 0.001 level. Note that Rinne series is only for the months MJJA and is reported under summer. Acronyms used for each series are given in Table 3.**

| IoI_1711 Series | Summary of key findings |
|---|---|
| Annual | IoI_1711 shows strong consistency with all other observational records throughout the series. There is thus high confidence in the annual series. The wettest decade in the series is the most recent (2006-2015), while the driest (1740-1749) is consistent with other long series covering this period. The early, independent, observations from Cork add confidence to the 1740s being exceptionally dry. |
| Winter (DJF) | IoI_1711 is consistent with other observational records from ~1790 onwards, increasing confidence in this part of the series. Prior to this there is divergence among all observational records. Confidence in the 1730s being an exceptionally wet decade in the IoI_1711 series is built from coherence with CET, L-NAO, WI, EU2 Index at this time. While dry, given coherence with a persistently negative L-NAO, the period from 1740-1780 is likely overly dry in our series. The driest decade in the IoI_1711 series (1777-1786) is broadly consistent with other records. |
| Spring (MAM) | IoI_1711 shows strong coherence with all observational records from 1740 onwards. Prior to this the number of comparison stations decreases (Spald/EWR), though consistency is maintained, thus increasing confidence in the entire series. There is also strong coherence with indicators of westerly flow throughout the record, with wet decades consistent with enhanced westerly flow, and dry spring conditions with reduced westerly flow. The wettest spring decade (1715-1724) is consistent with a strongly positive WI and the driest spring decade (1831-1840) is consistent with EWP and Oxford series. |
| Summer (JJA) | The key feature of the IoI_1711 summer series are the exceptionally wet decadal totals through the latter half of the 1700s. While this period is also exceptionally wet in other observational records and tree ring reconstructions, particularly in the 1750s and 1760s, the persistence of wet conditions is not evident in subsequent decades. Wet conditions throughout the period are consistent with a persistently positive L-NAO and WI, and low L-SLP and EU2 index. Otherwise, consistency with long-term observations increases confidence in the IoI_1711 summer series. |
| Autumn (SON) | IoI_1711 shows strong coherence with long term observations and models of variability throughout the record, thus building confidence in the entire series. The driest (1745-1754) and wettest (1770-79) decades are consistent with other long-term series, as is the notable variation in the early record. |

**Table 5 Synthesis of key findings from the comparison of IoI_1711 annual and seasonal decadal means with other long-term series.**

| Year | Seasonal Summary |
|---|---|
| 1730 | Spring variable. Summer wet. Autumn variable. Winter open, mild and comparatively dry. |
| 1731 | A dry and cold spring, but concluded hot. A hot and dry summer. Autumn variable and sometimes windy. Winter wet and warm. |
| 1732 | Variable weather in the spring, the last month rainy. Summer moderately fair and dry. Autumn wet and windy. A wet, windy, stormy and warm winter. |
| 1733 | A very dry spring. A dry summer but ended wet and windy. A wet and windy autumn. A wet windy and very warm winter, as so in England. The primroses and violets were blooming at Christmas. |
| 1734 | The two first moths of spring very warm (and so for England) but followed by a cold and nipping May, which hurted the fruits and burnt the grass. A wet summer: much straw and little grain: grass plentiful in the uplands. Autumn variable. A wet, windy and generally mild winter. N.B. the state of the weather in England from Short's Chronological History of the Weather and Seasons agrees very nearly to ours. " From September 28th 1731, to June 12th 1734 it was mostly droughty, no general or great floods or rains, and the springs failed in most places: then June 12th 1734, began the long wet season, and continued mostly to February 2nd 1736, viz a year and eight months, after two years and nine months drought" |
| 1735 | Of the spring the two first months were pretty open; the succeeded by a cold and dry May. The summer cold and wet like winter. Autumn wet. Winter open. Abundance of moisture through the three seasons of summer, autumn and winter. |
| 1736 | This summer was as remarkable for heat as the preceding one had been for coldness and moisture. A mild spring for the most part, though not without some frost and changeable weather. One of the hottest summers that has been remembered. Autumn moderately fair and mild. Winter very open, wet and windy: little frost. |
| 1737 | Spring warm, and May excessively hot. Summer mostly fair, but great changes as to the temperature of the air. Some days excessively hot, others very cold, and August a winter like month. Autumn fair and mild. Winter open. |
| 1738 | Spring seasonable. Summer cold and wet, except July which was hot and dry. Autumn for the most part wet. Winter wet and stormy. |
| 1739 | Spring mostly cold. Summer very wet. Autumn variable. Winter frosty, after a long series of open winters. |

**Table 6 Seasonal weather summaries for each year in the 1730s taken from the weather diary of Dr John Rutty (1770).**

| Year | Seasonal Summary |
|---|---|
| 1750 | Spring cold, dry, and backward. Summer (excepting a few excessively hot days) cold, moist, and winter-like. Autumn variable, mild at the beginning, frosty after. Winter (except December, which was warm) exhibited a good deal of frost and snow. |
| 1751 | Spring cold. Summer wet. Autumn variable. Winter hazy and cloudy, little frost. |
| 1752 | Spring cold and dry, excepting a moist May. Summer extremely wet. Autumn moderate and dry. Winter frosty, with snow and frequent rains. |
| 1753 | Spring seasonable, excepting a wet March. Summer wet, not above one half summer-like. Autumn fair and dry, ending frosty. Winter rainy, and great floods with frosts interposed. |
| 1754 | Spring partly temperate, partly cold. Summer wet. Autumn fair and summer-like. Winter frosty. |
| 1755 | Spring wet. Summer wet. Autumn wet, except October. Winter wet. |
| 1756 | Spring variable, a cold and moist April. Summer very wet. Autumn variable. Winter frosty. |
| 1757 | Spring cold and backward. Summer generally cloudy and wet, except June, and a few days in July and August. Autumn mostly dry and summer like. Winter mostly mild and open. |
| 1758 | Spring cold and dry. Summer rainy for the most part. Autumn mostly dry and fair. Winter mild. |
| 1759 | The spring mostly fair and dry. The summer mostly fair, dry and warm. The autumn mostly fair and moderate. The winter variable, but more inclined to moisture. |

**Table 7 Seasonal weather summaries for each year in the 1750s taken from the weather diary of Dr John Rutty (1770).**

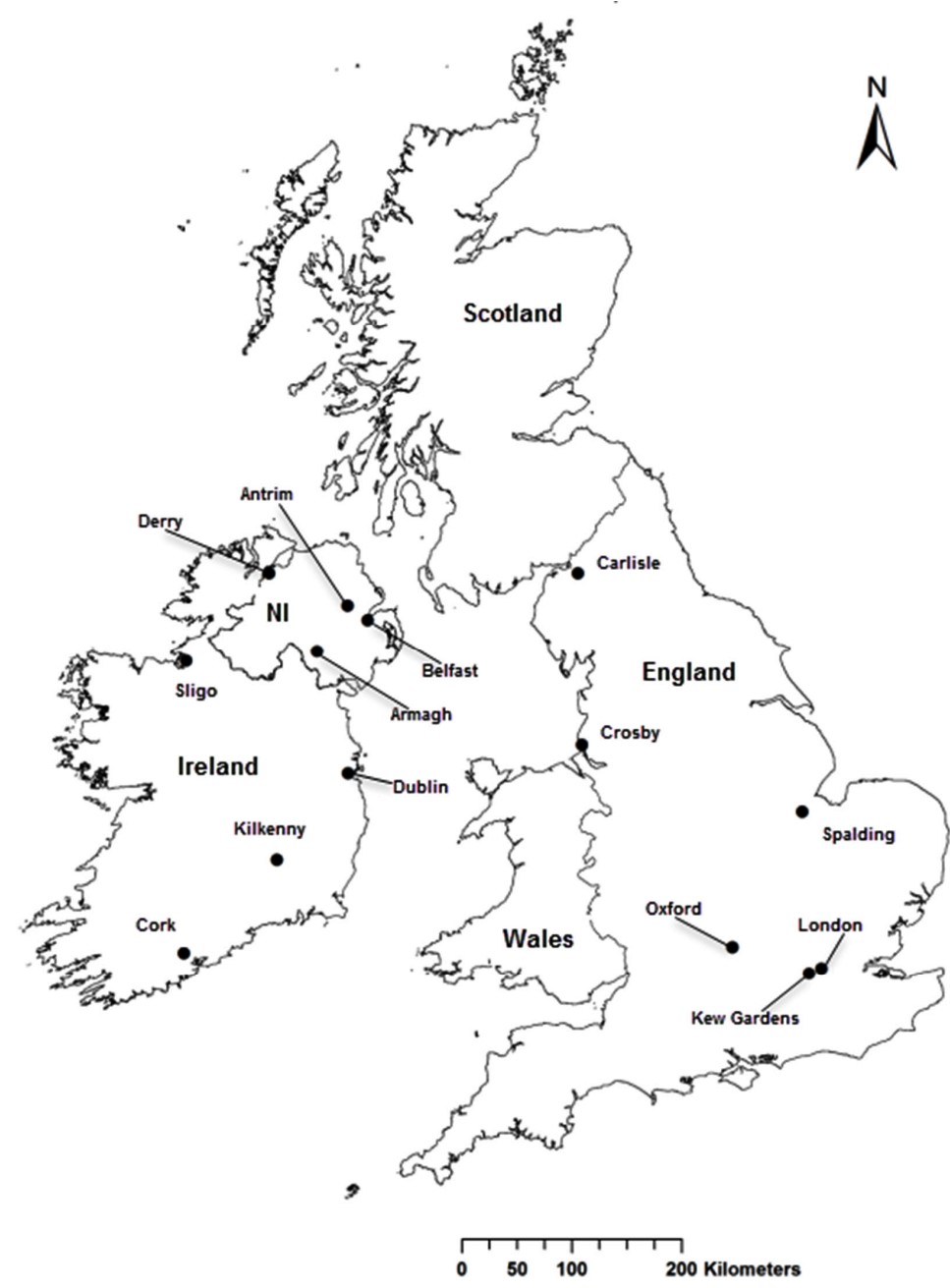

**Figure 1 Location of sources comprising the Jenkinson et al. (1979) data, together with locations of comparison precipitation series used in the analysis. Please note that in terms of the latter we also employ the Hoofddorp series (Amsterdam), which is not plotted on the map. Start dates of the various series can be found in Tables 1 and 2 for Jenkinson sources and comparison series, respectively.**

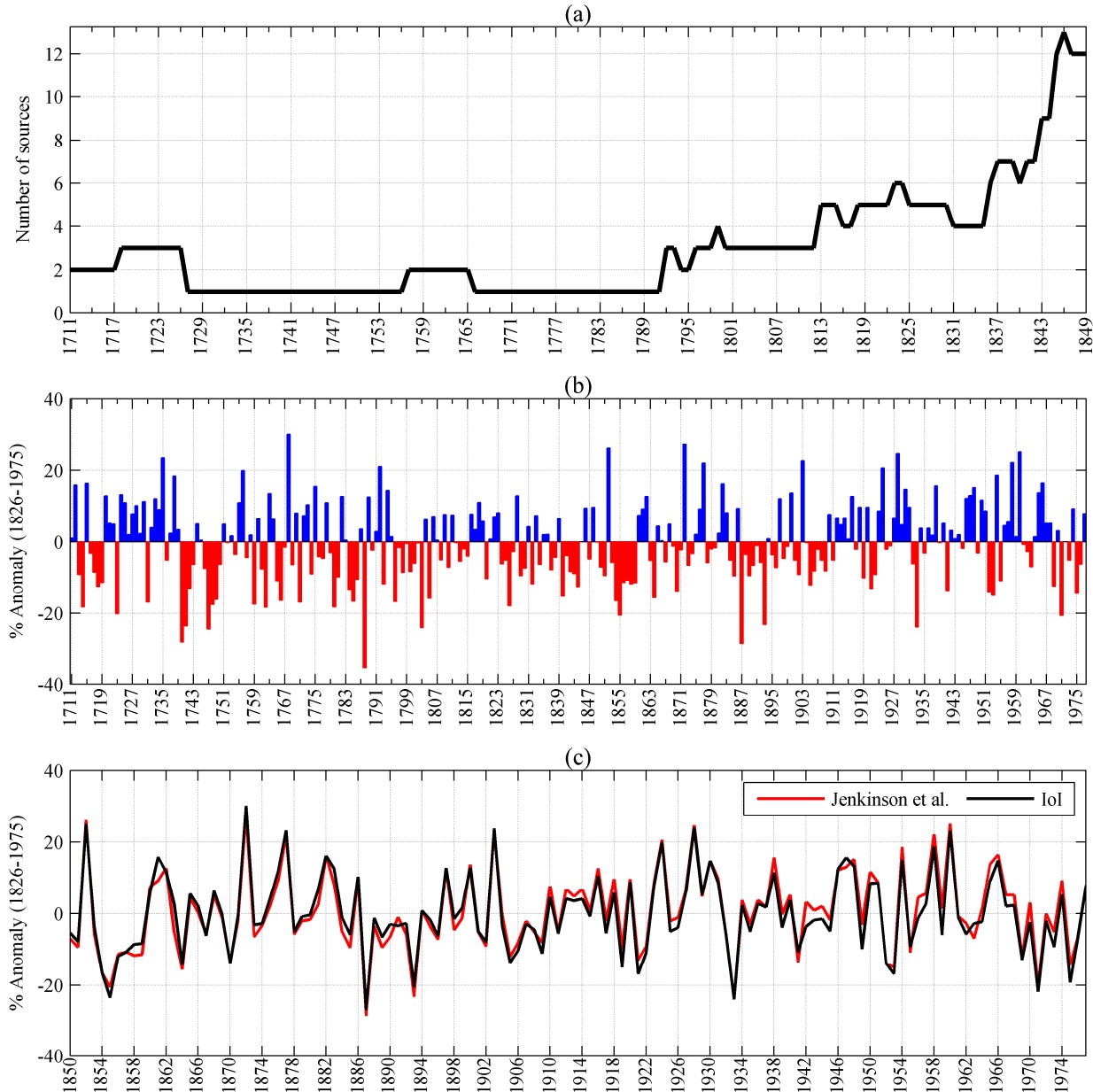

**Figure 2 a).** The number of constituent sources used to compile the Jenkinson data for the years 1711-1849; b). percentage annual rainfall anomalies (relative to the normal period 1826-1975) contained in Jenkinson et al. (1979) for the years 1711-1977 and c). comparison of b). with anomalies from IoI_1850 for years 1850-1977 (anomalies from 1850-1975 mean representing closest approximation of the Jenkinson et al. (1979) normal).

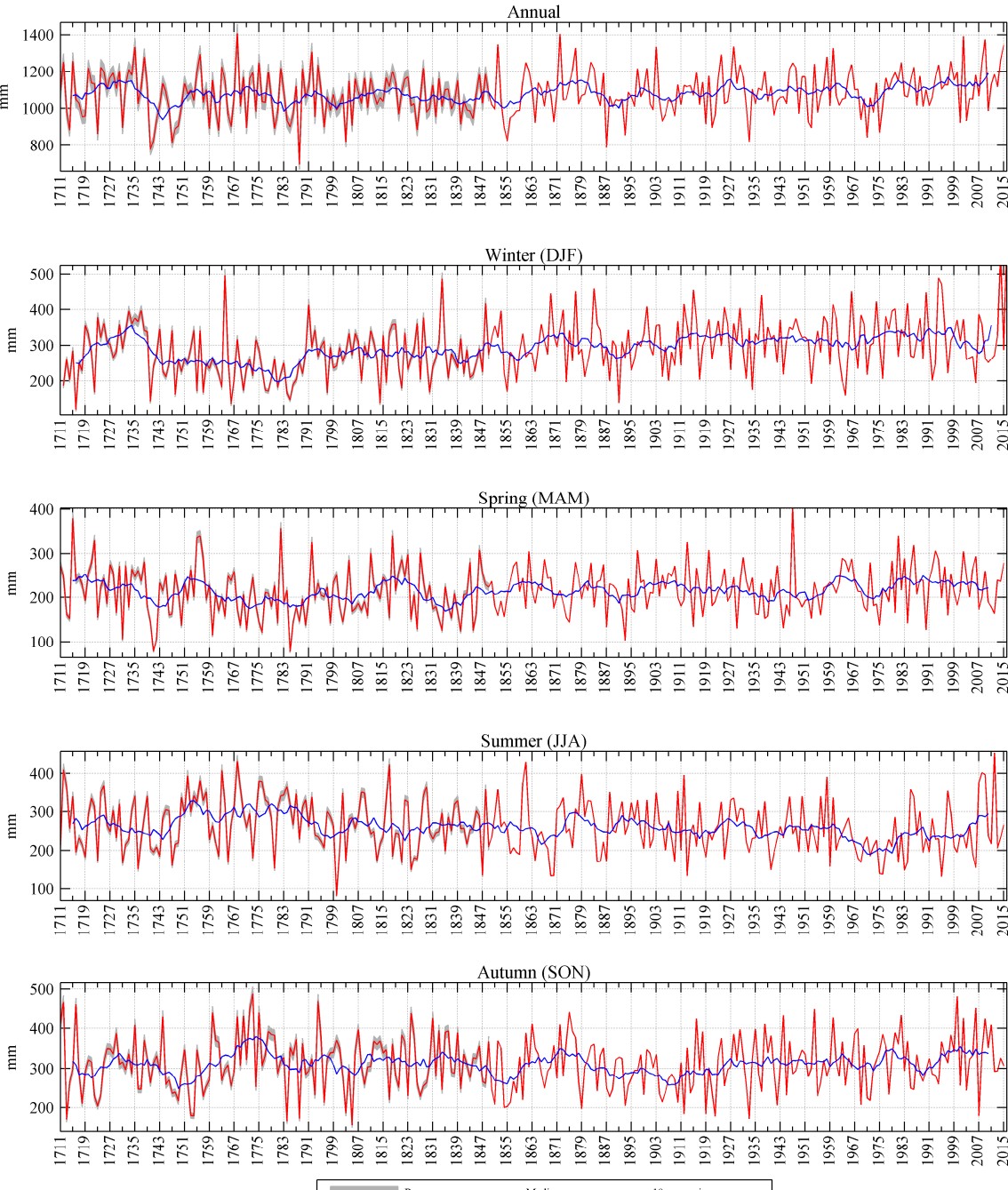

**Figure 3 Reconstructed IoI_1711 precipitation series showing annual and seasonal totals. The grey shading shows the uncertainty in the reconstruction from resampling of the baseline used to estimate AAR only. The red line (used in subsequent analysis) is the median of the 1000 resamples. From 1850 onwards the data is the IoI_1850 series produced by Noone et al. (2016). The blue line represents a 10-year moving average.**

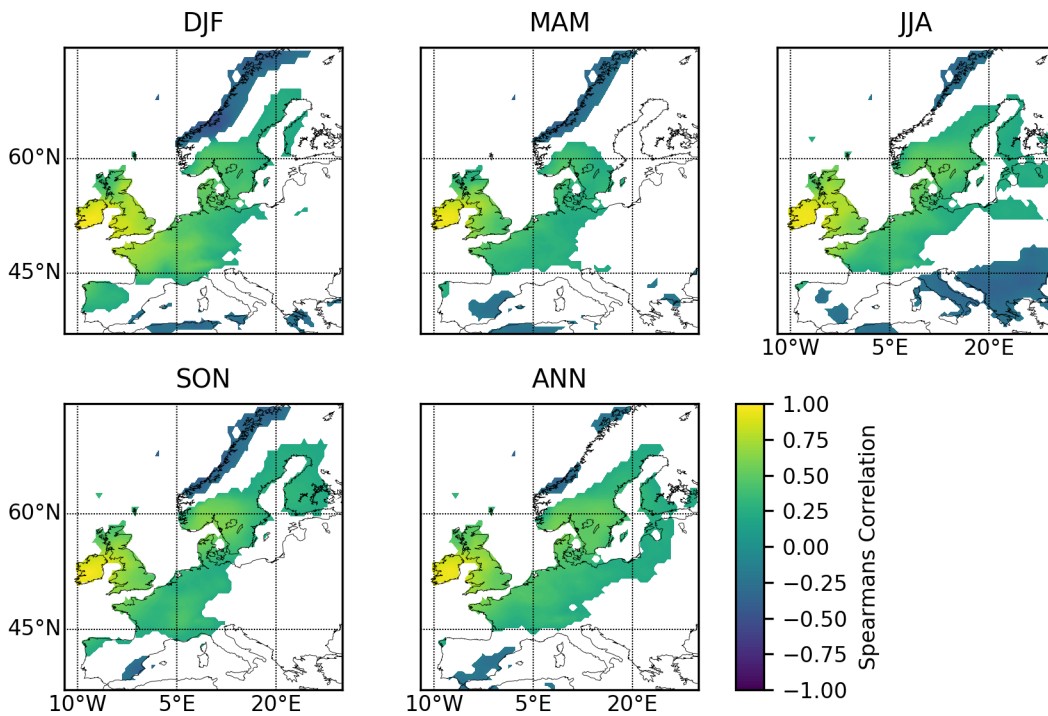

**Figure 4 Seasonal and annual Spearman's Rho correlations between IoI_1711 and gridded (0.5° resolution) CRU TS V4.1 precipitation for the years 1901/02 - 2015/16. Grids for which p > 0.05 are denoted white.**

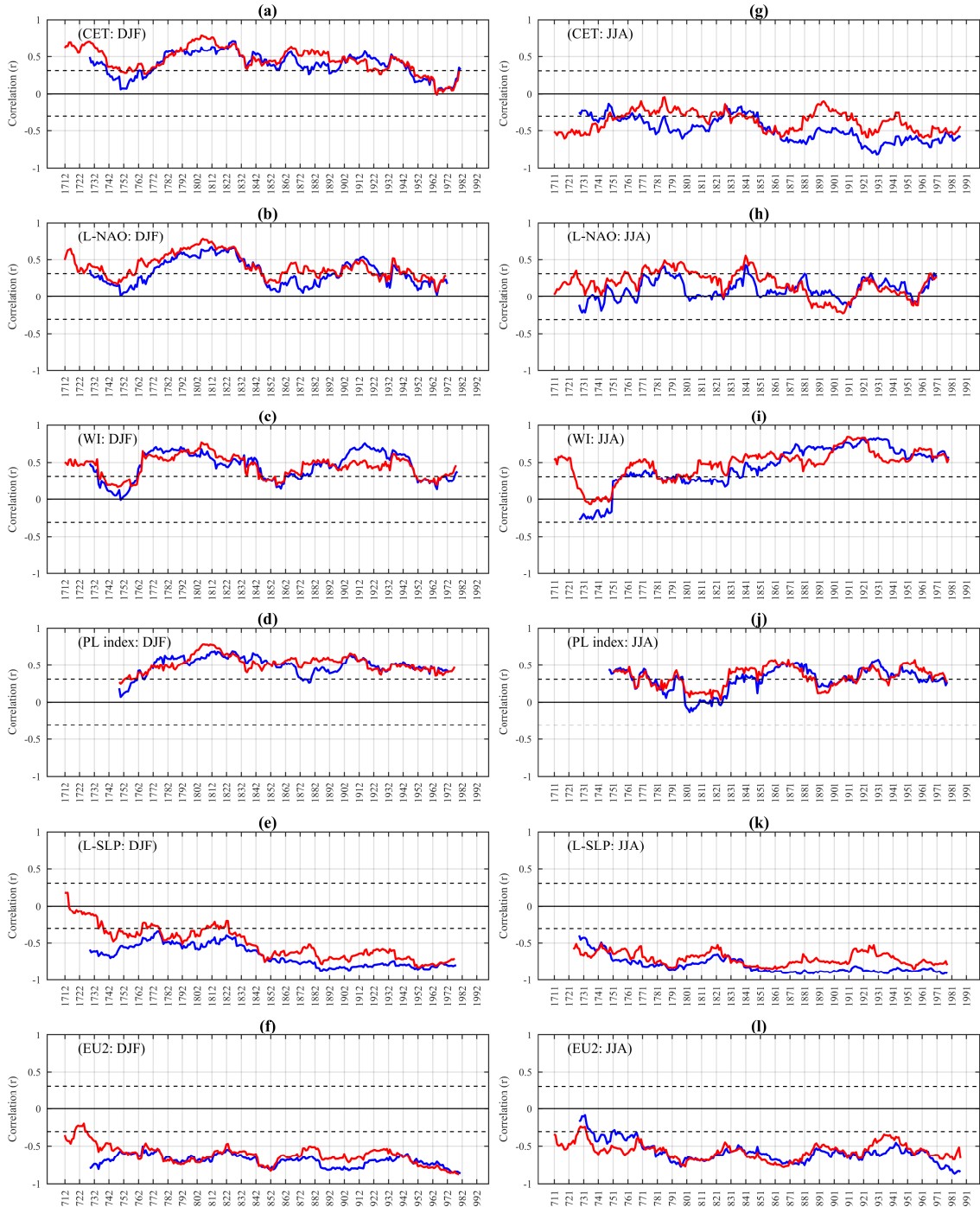

**Figure 5 Moving 30-year correlations between IoI_1711 (red) and EWR/EWP (blue) for winter (left) with a) CET; b) L-NAO; c) WI; d) PL index; e) L-SLP; f) EU2 index and for summer (right) with g) CET; h) L-NAO; i) WI; j) PL index; k) L-SLP; l) EU2 index. Acronyms used for each series are given in Table 3. Dashed lines indicate 95% confidence levels for moving correlations identified using a Monte Carlo procedure.**

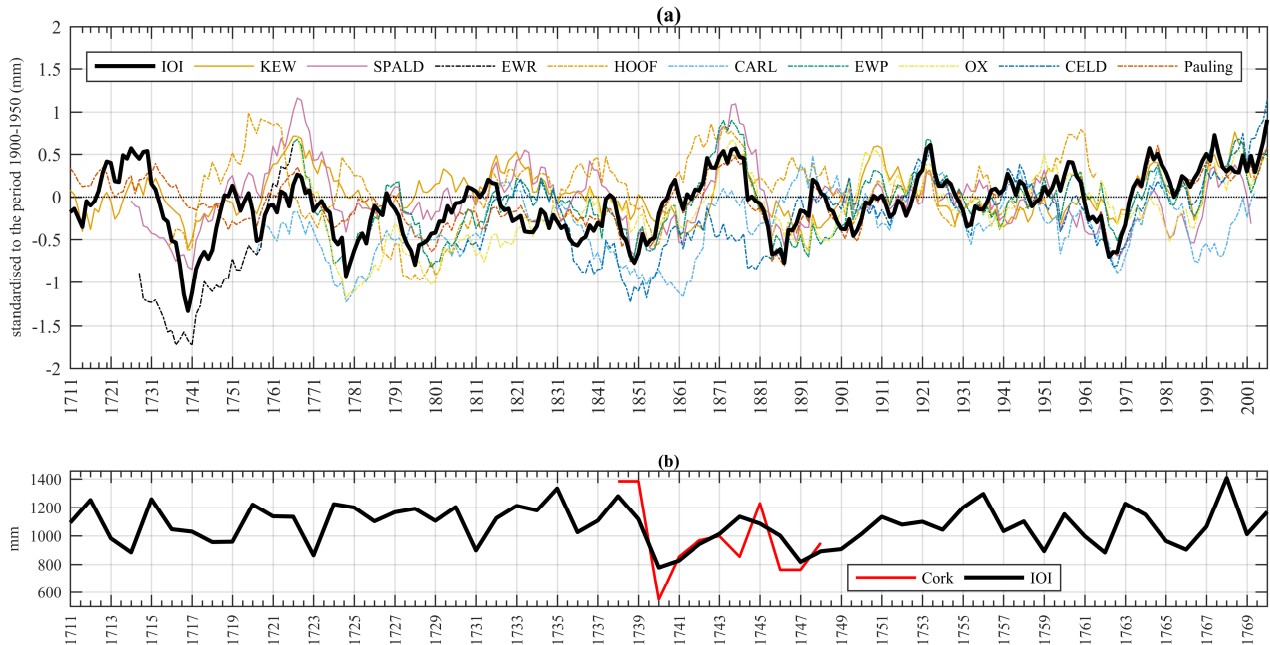

**Figure 6 a) Comparison of decadal mean IoI_1711 annual series (thick black line) with other long term precipitation records standardised to the period 1900-1950 and b) comparison of raw annual totals (mm) from Cork with annual totals from the IoI_1711 series. Acronyms used for each series are given in Table 3.**

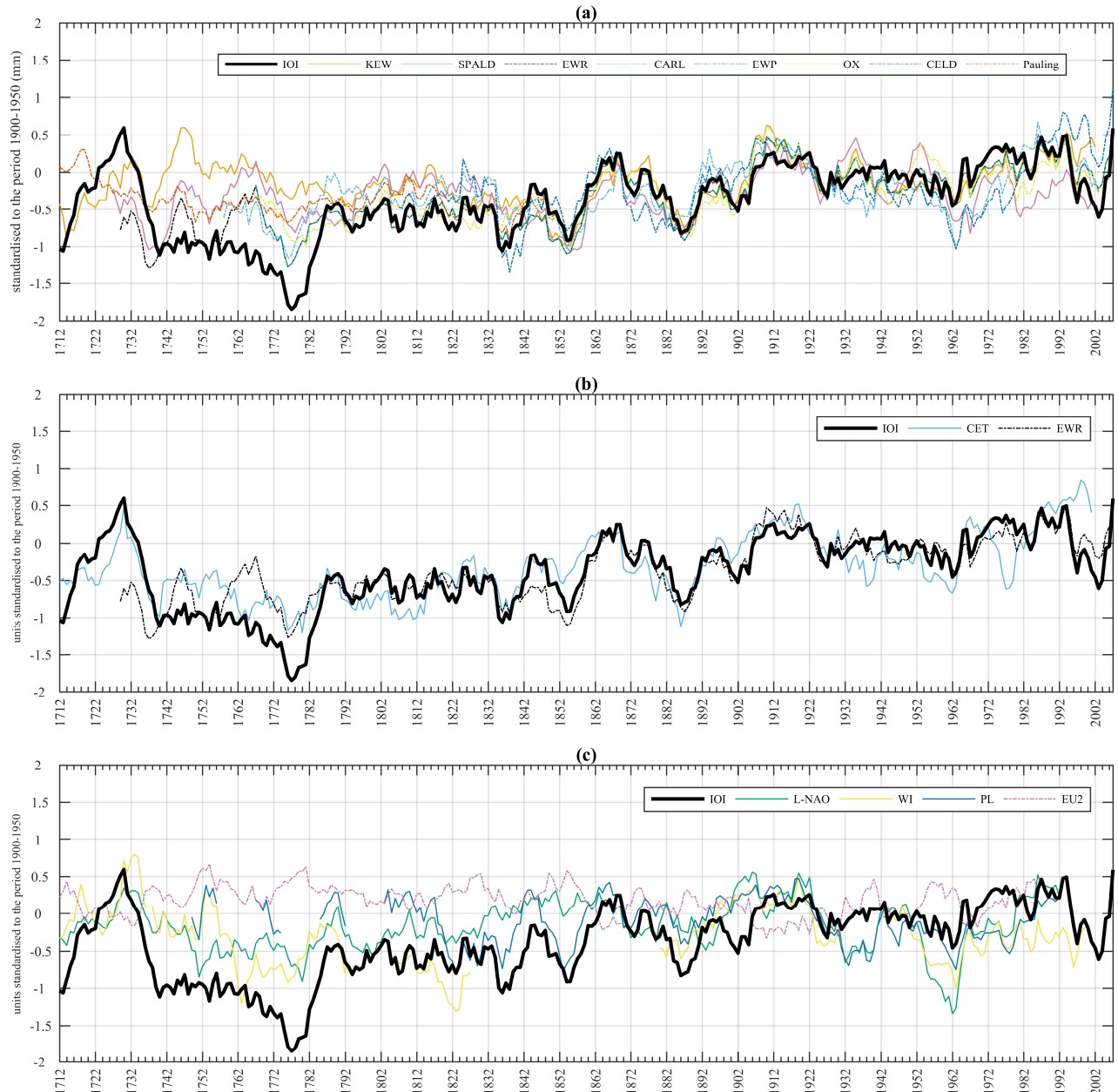

**Figure 7 Comparison of decadal mean IoI_1711 winter (DJF) series (thick black line) with a) other long term precipitation series; b) with CET, L-NAO and WI. In c) a comparison between the three indicators of westerly air flow (L-NAO, WI and PL index) is provided. All series are standardised to the period 1900-1950. Acronyms used for each series are given in Table 3.**

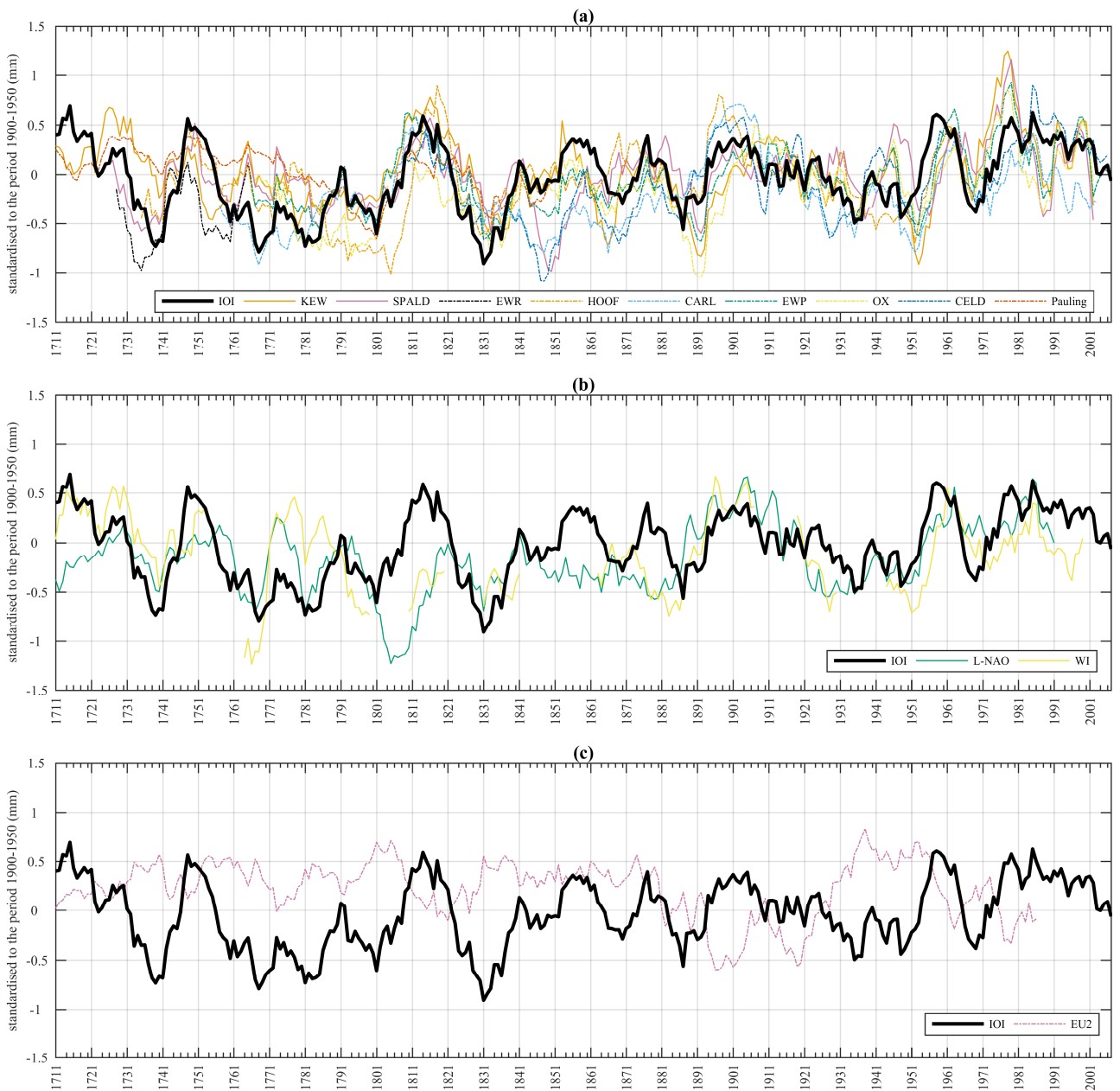

**Figure 8 Comparison of decadal mean IoI_1711 spring (MAM) totals (thick black line) with a) other long term precipitation series and b) indices of westerly airflow (L-NAO and WI). All data are standardised to 1900-1950. Acronyms used for each series are given in Table 3.**

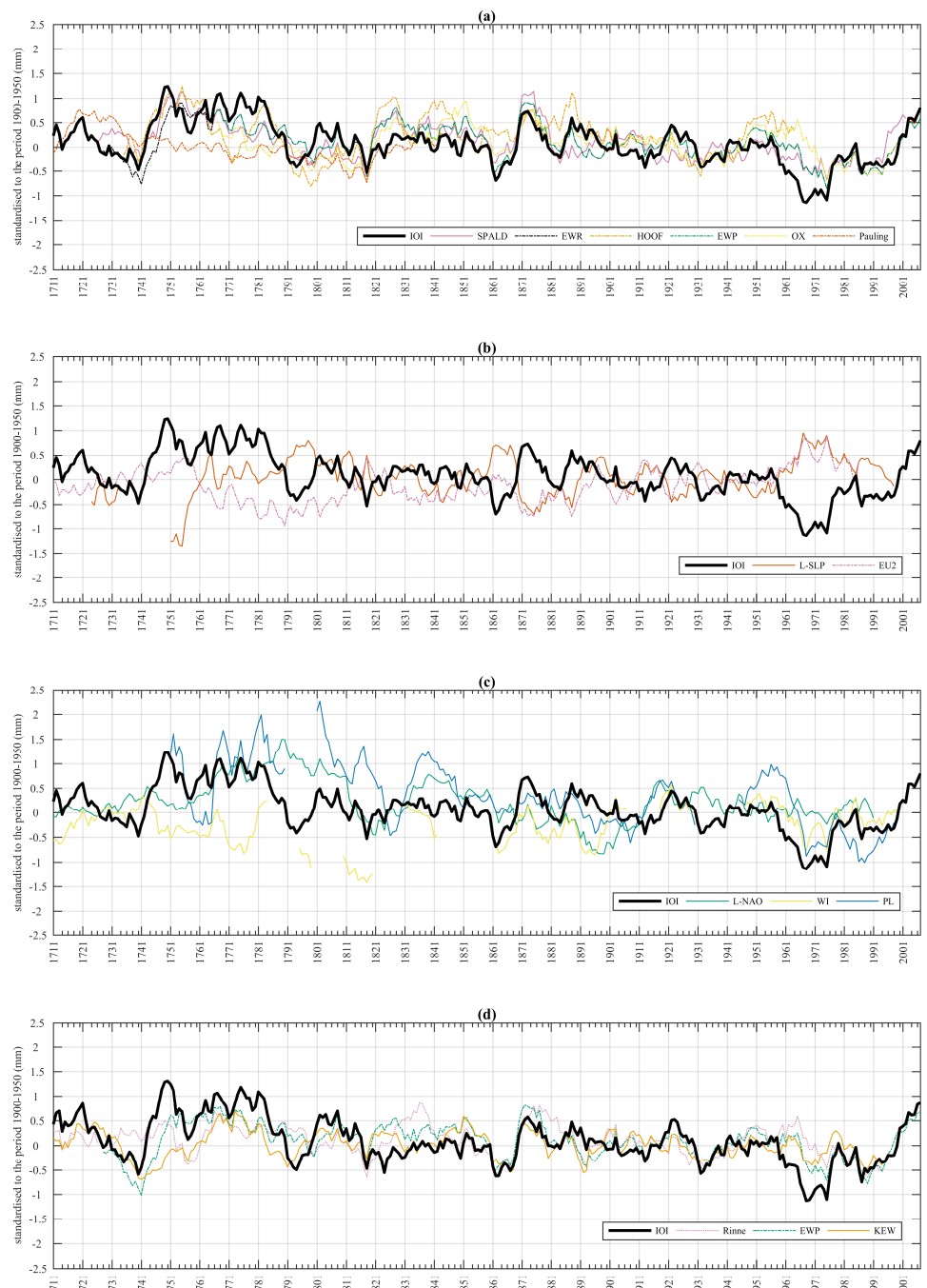

**Figure 9 Comparison of decadal mean IoI_1711 summer (JJA) precipitation totals with a) other long term precipitation series; b) with London Sea Level Pressure (L-SLP); c) with indicators of westerly flow (L-NAO, WI, PL index). Finally d) shows comparison of summer decadal means from IoI_1711 with tree ring reconstructions (Rinne) and EWP for the months MJJA. All data are standardised to 1900-1950. Acronyms used for each series are given in Table 3.**

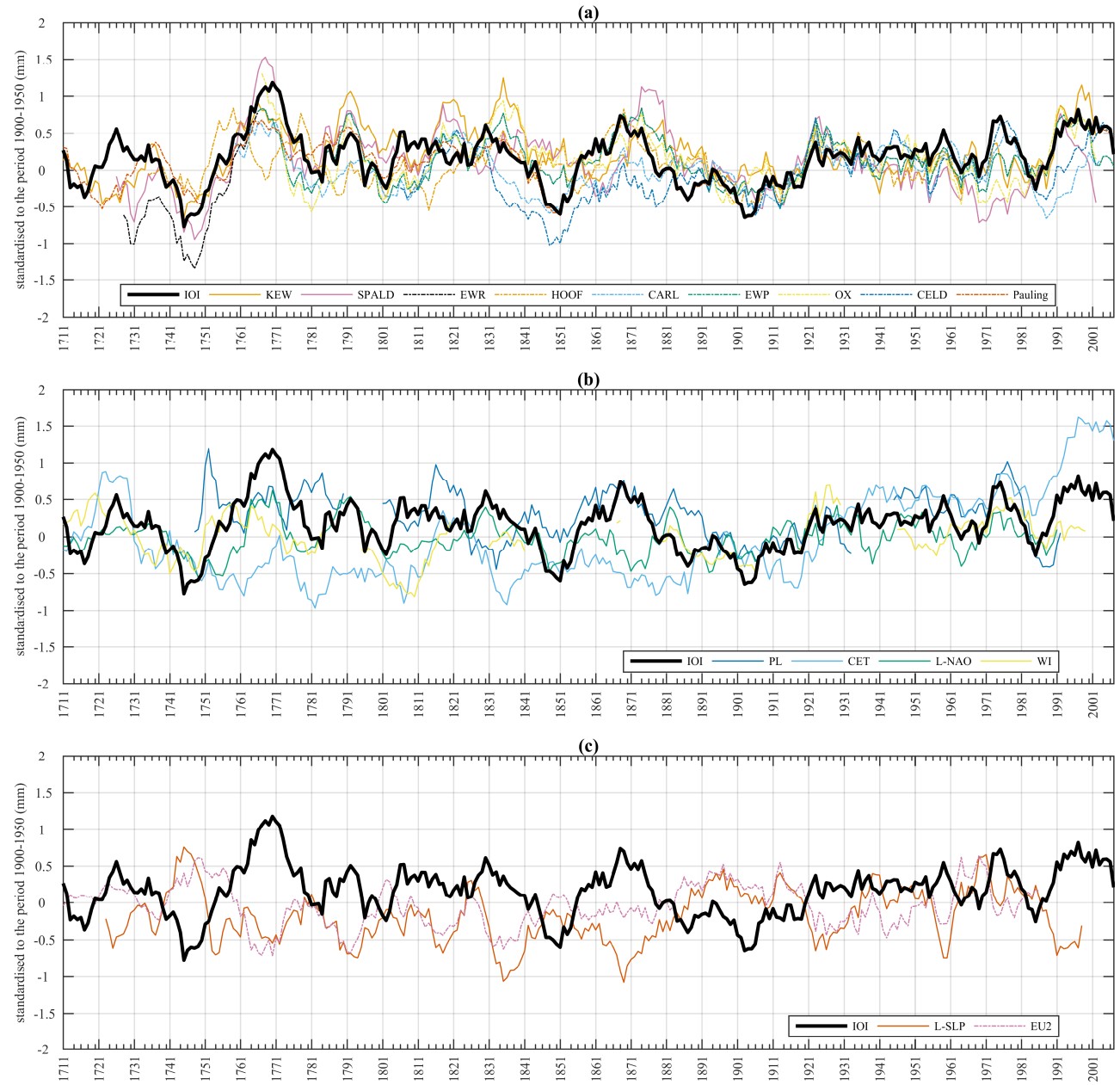

**Figure 10 Comparison of decadal mean IoI_1711 autumn (SON) precipitation totals with a) other long term precipitation series; b) with the Paris London Index and c) with London Sea Level Pressure. All data are standardised to 1900-1950. Acronyms used for each series are given in Table 3.**

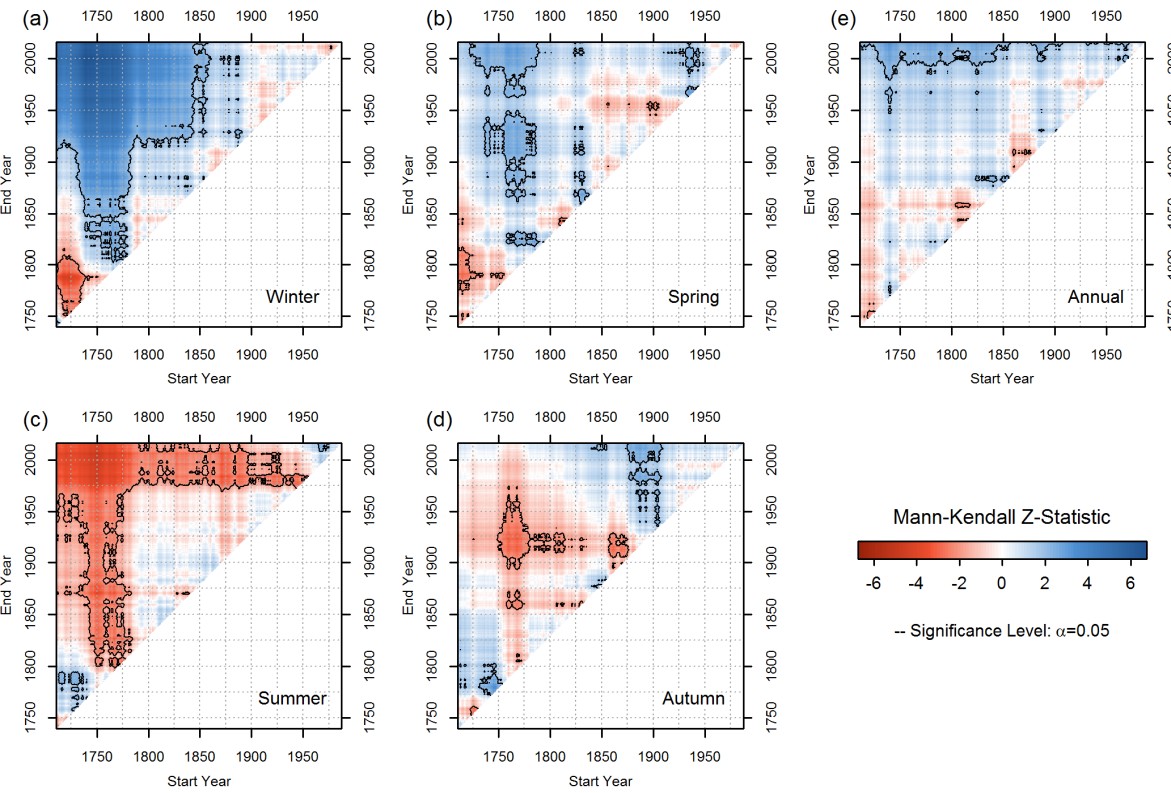

**Figure 11 Mann Kendall Zs values for IoI_1711 seasonal and annual series for all possible combinations of start and end years with minimum length of 30 years. Contours represent periods for which trends are significant at the 0.05 level.**

