# Peer review of "Met O 13 Branch Memorandum"

_Climate of the Past, 2017_

## Referee Comment (RC1) · J. Luterbacher (Referee) · 14 Nov 2017

General: This is an important contribution in deriving a continuous very long precipitation series for an area that is strongly influenced by the Atlantic and various circulation modes. The authors do a very good job in compiling all the meta information, the way how they generate the new series and provide associated uncertainties. They also carefully compare with other series and link statistically to upstream Atlantic climate conditions. I find this is an important contribution and I support it to be published in Clim Past. Below are a couple of suggestions/comments I wish the authors can take into consideration for the revisions.

Major comments:

[Figure]

Abstract: The abstract might be improved. In my view it is too detailed with specific information that do not reflect the major findings of the study. It might include a synthesis of the findings and what makes this series unique. Also implication of the results and potential for future applications could be stressed.

I suggest to provide a spatial correlation (spearman) plot between precipitation of Ireland (the representative station or country average) and Europe, separated for each season and for at least the last 50 years. That would show how the target new precipitation series is statistically linked to remote areas of the UK and European mainland.

I would suggest to produce a Figure that shows the location of the precipitation series that are compared with the target Ireland series. The starting year of the stations could be separated with different colors.

Section 3.2., table 4 and fig 3: Please could you also calculate the significance for the running correlation. One way would be through bootstrapping. For the interpretation of this figure I would then only describe and interpret the significant periods

You may need to state why you start in 1790

Comparison with independent circulation: The authors might use the independent gridded SLP (and derived NAO-I, first EOF of gridded SLP) reconstruction by Küttel et al. (2010). They cover the past 260 years and are fully independent.

Apart from the NAO, the East Atlantic/Western Russia pattern (Eurasia-2 pattern EU2 (Barnston and Livezey, 1987) is one of the most important modes for western Eurasia and shows also significant correlation with precipitation over Ireland. For instance, the EU2 index measures the pressure difference across central Europe and thus is important in describing the variability of Eurasian climate, especially during wintertime. Therefore the authors might also consider the EU index that has been reconstructed back to AD 1675 (Luterbacher et al. 1999) and calculate the correlation and running correlation.

It would also be interesting to compare your new reconstruction with those of Pauling et al. (2006) and Casty et al. (2007). The authors provide 0.5°x0.5° resolved gridded precipitation back to the mid 17th century (Pauling et al. 2006) and 1766 (Casty et al. 2007) including Ireland. The period could be analysed where there is no overlap of predictors. That would be another benchmark test for the various datasets.

Different studies have shown, that the AMO/AMV are significantly correlated with precipitation/temperature downstream over the UK and Europe. It would be interesting to see whether the positive correlation within the instrumental period is also valid during the reconstruction period. There is new annual AMO/AMV reconstruction available that could be used (Wang et al. 2017). The data can be downloaded from this link https://www1.ncdc.noaa.gov/pub/data/paleo/reconstructions/wang2017/wang2017amv-amo.txt

It is striking, that the L-SLP is the only index that shows a significant negative correlation. Why is this?

Decadal interpretations I suggest you also consider the Casty et al. (2007) and Küttel et al. (2011) papers with respect to linkages between the large scale atmospheric circulation and precipitation in Europe back to the mid-18th century.

It would help the reader if the seasonal aspects and the comparison with other series could be shortly summarized, that would help the reader keeping the major features from the many numbers. This could be also in the form of a table that synthesis the results.

Minor comments: Page 6, top: Jenkinson et al. (1979) applied a graded scaling system, similar to Brázdil et al. (2010), to both diaries.

In this context, please also cite Gimmi et al. (2007)

Please note that Luterbacher et al. 2002 was published in 2001 and the publication date should be changed accordingly.

Page 11: All records were standardised (by mean and standard deviation) to the period 1900-1950 for visual comparison. Could you please specify why this period and if another more recent period would be chosen, if the results would be stable? Why only visually? What is the intention?

Concerning the precipitation conditions in 1816 in Ireland, please have a look at Veale and Endfield (2016) with additional information

Concerning the post precipitation conditions in Ireland after the Laki eruption, please have also a look at Brazdil et al. 2010

Potentially the following paper is of relevance and could be included/cited

Evaluating the Dendrochronological Potential of Taxus Baccata (Yew) in southwest Ireland (Galvin et al. 2014)

The authors use Maunder Minimum as a key word but only mention it once. I would thus remove it.

  References used:

Barnston, A. G., and R. E. Livezey, 1987: Classification, seasonality and persistence of low frequency atmospheric circulation patterns, Mon.. Wea. Rev., 115, 1083-1126.

Brázdil, R., Demarée, G.R., Deutsch, M., Garnier, E., Kiss, A., Kolář, P., Luterbacher, J., Macdonald, N., and Rohr, C., 2010: Floods of the winter 1783/1784 in Europe: a scenario of an extreme event in the Little Ice Age, Theor. Appl. Climatol. 100, 163-189.

Casty, C., Raible, C.C., Stocker, T.F., Wanner, H., and Luterbacher, J., 2007: European climate pattern variability since 1766, Clim. Dyn., 29, 791-805.

Galvin, S.., A.P. Potito and K.R. Hickey (2014) 'Evaluating the Dendrochronological Potential of Taxus Baccata (Yew) in southwest Ireland'. Dendrochronologia, 32:144-152

[Figure]

Gimmi, U., Luterbacher, J., Pfister, C., and Wanner, H., 2007: A method to reconstruct long precipitation series using systematic descriptive observations in weather diaries: the example of the precipitation series for Bern, Switzerland (1760-2003). Theor. Appl. Climatol., 87, 185-197.

Küttel, M., Xoplaki, E., Gallego, D., Luterbacher, J., Garcia-Herrera R., Allan, R., Barriendos, M., Jones, P.D., Wheeler, D., and Wanner, H., 2010: The importance of ship log data: reconstructing North Atlantic, European and Mediterranean sea level pressure fields back to 1750. Clim. Dyn. 34, 1115–1128.

Küttel, M., Luterbacher, J., and Wanner, H., 2011: Multidecadal changes in winter circulation-climate relationship in Europe: frequency variations, within-type modifications, and long-term trends. Clim. Dyn., 36, 957–972.

Luterbacher, J., Schmutz, C., Gyalistras, D., Xoplaki, E., and Wanner, H., 1999: Reconstruction of monthly NAO and EU indices back to AD 1675. Geophys. Res. Lett., 26, 2745-2748.

Pauling, A., Luterbacher, J., Casty, C., and Wanner, H., 2006: 500 years of gridded high-resolution precipitation reconstructions over Europe and the connection to large-scale circulation, Clim. Dyn. 26, 387-405.

Veale, L and Endfield, GH (2016) Situating 1816, the 'year without summer', in the UK. The Geographical Journal, 182 (4). 318 - 330.

Wang J, Yang B, Ljungqvist FC, Luterbacher J, Osborn TJ, Briffa KR and Zorita E (2017) Internal and external forcing of multidecadal Atlantic climate variability over the past 1,200 years. Nature Geoscience 10, 512-517

Please also note the supplement to this comment:
https://www.clim-past-discuss.net/cp-2017-142/cp-2017-142-RC1-supplement.pdf

---

## Referee Comment (RC2) · L. Ashcroft (Referee) · 6 Dec 2017

Review of Murphy et al 2017

This paper describes an extension of a recent rainfall dataset for the Island of Ireland, taking the existing 1850–2016 dataset back to 1711 using a recovered summary found at the Irish Meteorological Service Met Eireann.

The study is one of those great data rescue stories that shows the value of this work. The data are well described, with limitations and caveats presented appropriately. I particularly liked the use of bootstrapping to get a range of estimates for the annual average rainfall used to construct the full series, and the visualisation of trends.

It's a shame that the authors can't find more information about how the recovered

summary was developed, but their use of independent and somewhat related sources to verify the extended record is well done. The information about other long-term rainfall data in and around Ireland in this paper will also surely be useful for future researchers.

I recommend that the article be published subject to the consideration of the small suggestions below.

- It would be good to include a map of Ireland, and perhaps the wider UK region, showing the locations mentioned in the text (including the independent sources) for those unfamiliar with Ireland.

- The study talks more about the rainfall variability in summer and winter than that of autumn and spring, despite the fact that the winter record is arguably the least reliable in the first few decades. Is there a reason for this? Are these key seasons for water security in Ireland, or seasons that show particular sensitivity to large-scale features?

- Using bootstrapping to estimate the AAR (and uncertainty) seems like a good idea to me. However, a reader might think that the uncertainty shown in Figure 2 represents all of the associated uncertainties with the data, including quality issues of the very early observations. You discuss this towards the end of the paper, but it would be good to add a disclaimer about this in Section 2.1.2 (and in the caption to Figure 2) to clarify.

- Is there any reason why you used Spearman rank correlation over Pearson?

- In section 2.2.1, you mention that the EWR series was used to calibrate the Jenkinson series. Can you elaborate on that? Did you do that, or Jenkinson et al?

- You mention where you obtained the Hoofddorp dataset, but no other comparison series. Are they all from the associated publications? Perhaps specify this.

- Section 2.3: did you do any quality control/outlier analysis on the data before homogenisation?

- Why did you use SNHT and the Pettitt tests for homogeneity assessment? Presumably it's because they don't need neighbouring stations, but it would be worth adding a sentence explaining your choice of these methods over more recent approaches such as RHtestV4.

-It's great that the original data source is provided with the paper, but will the final digitised dataset also be made available?

- A hopeful question: have you looked into whether Jenkinson or their colleagues are still alive?

- As an aside: this work, while suitable for Climate of the Past, would also have been suitable for data-focussed publications such as Geoscience Data Journal or Earth System Science Data. This would have enabled you to attach a DOI to the Jenkinson record (and your dataset) and make it more prominent, rather than including it as an attachment.

-In section 3.3.1 you mention that the post 1950 period is the wettest for winter, but in the discussion talk about the impact that increasing availability of observations has on the trend. Do you think that this post-1950 wet signal might also be an indicator of increasing data coverage?

Table 4: Two decimal places in the correlations is probably enough

Table 5 and 6: One caption mentions that these descriptions are 'derived from', while the other says 'taken from' Rutty's diary. Were they compiled in different ways?

Figures 4 to 9: You might want to try some different colour schemes for these spaghetti-type plots that are colour-blind appropriate (try http://colorbrewer2.org/). the IoI curve should ideally by on top of the other too, I think that would improve readability. Finally, it would be great if you could spell out the acronyms used in the legend, at least in the first caption.

Technical corrections

Abstract, line 10: I'd add the word 'boreal' before 'spring, summer and autumn', for southern hemisphere readers.

Abstract, line 14: add the word 'volcanic' before eruption

Page 3, line 8: do you have a reference about the lost diaries of William and Sam Molyneux?

Page 4, line 11: I'd add the word 'precipitation' before 'record'.

Page 4, line 21: I'd add 'community standard' or something similar ahead of the mention of HOMER, to signify its standing in the homogenisation field

Page 11, line 17: I feel like a word is missing at the start of this sentence. Maybe 'To derive the IoI_1711 series MEAN'?

Page 11, line 23: you talk about the median and mean of the series in the same sentence, is that accurate?

Page 12, line 28: I'd remove the 'For' at the start of this sentence

Page 17, line 35: Maybe add 'around that time' to the end of this sentence, and include a reference to George J Symons' network if you have one

Page 19, line 2: remove 'multiple' or 'different'

Page 19, line 5: add 'previously' before 'available'

Page 19, line 26: 'remain', rather than 'remains'

---

## Short Comment (SC1) · 6 Dec 2017

Very nice and interesting paper, with a very important time series. I think the Royal Navy had a big presence at Cork starting around the late 1700s, and perhaps later work could incorporate the original ship logs from there.

---

## Author Comment (AC1) · 12 Dec 2017

Many thanks indeed for your kind words and valuable feedback. You are right. Tyrrell (1995) did some interesting work with navy log books, but I am sure there is more that could be done here. We will make note of this potential avenue in our revisions.

Tyrrell, J. G.: Paraclimatic statistics and the study of climate change: The case of the Cork region in the 1750s, Clim. Change, 29, 231–245, 1995.

---

## Author Comment (AC2) · 10 Jan 2018

**Reviewer 1 – Prof. Luterbacher**

This is an important contribution in deriving a continuous very long precipitation series for an area that is strongly influenced by the Atlantic and various circulation modes. The authors do a very good job in compiling all the meta information, the way how they generate the new series and provide associated uncertainties. They also carefully compare with other series and link statistically to upstream Atlantic climate conditions. I find this is an important contribution and I support it to be published in Clim Past. Below are a couple of suggestions/comments I wish the authors can take into consideration for the revisions.

We thank Prof. Luterbacher for the time taken to provide such a constructive review and for the positive comments above.

Abstract: The abstract might be improved. In my view it is too detailed with specific information that do not reflect the major findings of the study. It might include a synthesis of the findings and what makes this series unique. Also implication of the results and potential for future applications could be stressed.

In our revisions we will rework the abstract as requested.

I suggest to provide a spatial correlation (spearman) plot between precipitation of Ireland (the representative station or country average) and Europe, separated for each season and for at least the last 50 years. That would show how the target new precipitation series is statistically linked to remote areas of the UK and European mainland.

An excellent suggestion that we hope will increase the uptake of our series in wider European work. We will include an additional figure and associated text around interpretation in the revised manuscript.

I would suggest to produce a Figure that shows the location of the precipitation series that are compared with the target Ireland series. The starting year of the stations could be separated with different colors.

This was a point raised by both reviewers and we have plotted the locations of comparison series across the UK. Reviewer 2 also asked for locations from Ireland mentioned in the text to be included so that readers not familiar with the geography of the island can get a feel for the data/locations. The revised paper will include an additional figure for this map. One challenge however is that given the number of locations now included and the multitude of start dates it is very difficult to include start year by colour. This information is given in the tables (Table 1 and Table 2) in the manuscript and we argue that this is sufficient.

Section 3.2., table 4 and fig 3: Please could you also calculate the significance for the running correlation. One way would be through bootstrapping. For the interpretation of this figure I would then only describe and interpret the significant periods.

Sure, we can do this for series in Figure 3. We will have a go at implementing the procedure in Pauling et al. (2006) for examining significance of running correlations. See additional comments on Fig. 3 below.

Comparison with independent circulation: The authors might use the independent gridded SLP (and derived NAO-I, first EOF of gridded SLP) reconstruction by Küttel et al. (2010). They cover the past 260 years and are fully independent.

Thank you for this recommendation. The London SLP series, the Paris London Index and the Westerly Index also provide independent insights into SLP and the NAO-I. Given the number of series we already incorporate we examined the Küttel et al. (2010) dataset to determine if it was consistent with the aforementioned series and the extent to which it could add value to the paper (given the series already included). We focus attention on the derived NAO-I (leading EOF of the gridded Küttel et al. SLP data).

Agreement between the leading EOF for each season and the Hurrel PC based NAO-I is very strong for winter and spring (r>0.9), but less so for summer and autumn. This is most likely because the domain used in the Küttel et al. SLP series extends further east (to 50 deg) and includes the Arabian heat low (which will be very prominent in summer/autumn).

Comparison of the winter (DJF) leading EOF shows a strong correlation with other modes of westerly flow already analysed in the paper. The table below shows Spearman's rank correlation matrix from 1760 – 2000 for winter (DJF) IoI rainfall and indices of westerly flow from our manuscript.

|  | IoI 1711 | WI | PL Index | L-NAO | Küttel et al. EOF |
|---|---|---|---|---|---|
| IoI_1711 | 1.00 | 0.50 | 0.31 | 0.46 | 0.32 |
| WI | 0.50 | 1.00 | 0.52 | 0.43 | 0.56 |
| PL Index | 0.31 | 0.52 | 1.00 | 0.52 | 0.67 |
| L-NAO | 0.46 | 0.43 | 0.52 | 1.00 | 0.71 |
| Küttel et al. EOF | 0.32 | 0.56 | 0.67 | 0.71 | 1.00 |

[Figure]

Evident is the strong correlation between Küttel et al. and the PL-index and L-NAO index. Comparison of decadal means (standardised to period 1900-1950, as in our manuscript) also shows the strong coherence between the leading winter EOF from Küttel et al. and the PL Index and L-NAO index. We therefore conclude that integrating the Küttel et al. data into the paper adds little value above the series already included. Given the number of series being plotted already we would therefore rather not include it to maintain legibility of the plots. In addition, the other series, including an, independent, homogenised SLP series for London, together with three estimates of westerly flow and the NAO-I, extend further back into the 1700s.

We will however mention in the discussion that this is a source that could have been used or may be useful in other similar work and highlight the consistency with other series as above.

Apart from the NAO, the East Atlantic/Western Russia pattern (Eurasia-2 pattern EU2 (Barnston and Livezey, 1987) is one of the most important modes for western Eurasia and shows also significant correlation with precipitation over Ireland. For instance, the EU2 index measures the pressure difference across central Europe and thus is important in describing the variability of Eurasian climate, especially during wintertime. Therefore the authors might also consider the EU index that has been reconstructed back to AD 1675 (Luterbacher et al. 1999) and calculate the correlation and running correlation.

This is an excellent suggestion. The EU2 index is indeed highly correlated with our 1711 series in all seasons, especially in winter. This is a valuable index to our study and we will integrate it into the paper in our revisions. Specifically this will now be included as one of our independent comparison series and integrated into the assessment of running correlations (Fig 3), the correlations in table 2 and each of the panel plots for the seasonal comparisons. Thanks for bringing this to our attention.

It would also be interesting to compare your new reconstruction with those of Pauling et al. (2006) and Casty et al. (2007). The authors provide 0.5°x0.5° resolved gridded precipitation back to the mid 17th century (Pauling et al. 2006) and 1766 (Casty et al. 2007) including Ireland. The period could be analysed where there is no overlap of predictors. That would be another benchmark test for the various datasets.

Again, an excellent suggestion. We have used the 1766 (Casty et al. 2007) series in previous work exploring historical droughts in Ireland and have found it particularly useful (Noone et al. 2017) However for this study, both in terms of length of record and independence of the early series the seasonal reconstructions by Pauling et al. (2006) are likely to be most useful. Stations in Ireland are used as predictors in this series, however, the earliest Irish precipitation source seems to be Armagh Observatory which extends to the 1830s. Prior to this the Pauling et al. data would provide a useful additional comparison series, though there remain issues of circularity which are impossible to avoid. In particular the Jenkinson data was calibrated to EWR by the original authors, with some of the UK series that comprise Pauling et al., also likely to be used for this purpose. Nonetheless we will include the Pauling et al. data as a series in our plots comparing running decadal totals with other precipitation series (Fig 4- Fig 8).

Different studies have shown, that the AMO/AMV are significantly correlated with precipitation/temperature downstream over the UK and Europe. It would be interesting to see whether the positive correlation within the instrumental period is also valid during the

reconstruction period. There is new annual AMO/AMV reconstruction available that could be used (Wang et al. 2017). The data can be downloaded from this link https://www1.ncdc.noaa.gov/pub/data/paleo/reconstructions/wang2017/wang2017amvamo.txt

Thanks for this suggestion. We will examine and explore this as part of our revisions.

It is striking, that the L-SLP is the only index that shows a significant negative correlation. Why is this?

In examining this comment we noticed that the running correlation plot (Fig 3) was wrong in our original manuscript. It repeats each plot for the same series and some other small issues. The correct plot is below, which we will of course update in our revisions (with the addition of the EU2 index). Significant negative correlations are evident between CET and IoI/EWR rainfall in summer and with L-SLP in both seasons. In relation to SLP, this is due to blocking – high SLP over London is associated with dry conditions over the islands. This is as expected. We would also expect the other indices to be positively correlated.

[Figure]

**Corrected Fig3 Moving 30-year correlations between IoI_1711 (blue) and EWR/EWP (red) for winter (left) with a) CET; b) L-NAO; c) WI; d) PL index; e) L-SLP and for summer (right) with f) CET; g) L-NAO; h) WI; i) PL index; j) L-SLP.**

Decadal interpretations I suggest you also consider the Casty et al. (2007) and Küttel et al. (2011) papers with respect to linkages between the large scale atmospheric circulation and precipitation in Europe back to the mid-18th century.

We will do so in our revised manuscript.

It would help the reader if the seasonal aspects and the comparison with other series could be shortly summarized, that would help the reader keeping the major features from the many numbers. This could be also in the form of a table that synthesis the results.

We will do this through either providing a short recap paragraph at the start of the discussion or a synthesis table in the results section.

**Minor comments**

Page 6, top: Jenkinson et al. (1979) applied a graded scaling system, similar to Brázdil et al. (2010), to both diaries. In this context, please also cite Gimmi et al. (2007)

We will do so

Please note that Luterbacher et al. 2002 was published in 2001 and the publication date should be changed accordingly.

We will do so

All records were standardised (by mean and standard deviation) to the period 1900-1950 for visual comparison. Could you please specify why this period and if another more recent period would be chosen, if the results would be stable? Why only visually? What is the intention?

We choose this period given the availability of data across all series, together with the good quality of observations from the period. The results do remain valid for different periods for standardisation. We will remove the reference to 'visually', this was included given we are plotting multiple series on the same axis.

Concerning the precipitation conditions in 1816 in Ireland, please have a look at Veale and Endfield (2016) with additional information

Thanks, we will integrate.

Concerning the post precipitation conditions in Ireland after the Laki eruption, please have also a look at Brazdil et al. 2010

Thanks, we will integrate.

Potentially the following paper is of relevance and could be included/cited Evaluating the Dendrochronological Potential of Taxus Baccata (Yew) in southwest Ireland (Galvin et al. 2014)

Thanks, we will integrate.

The authors use Maunder Minimum as a key word but only mention it once. I would thus remove it.

Will remove as a key word.

References in responses

Casty, C., Raible, C.C., Stocker, T.F., Wanner, H., and Luterbacher, J., 2007: European climate pattern variability since 1766, Clim. Dyn., 29, 791-805.

Küttel, M., Xoplaki, E., Gallego, D., Luterbacher, J., Garcia-Herrera R., Allan, R., Barriendos, M., Jones, P.D., Wheeler, D., and Wanner, H., 2010: The importance of ship log data: reconstructing North Atlantic, European and Mediterranean sea level pressure fields back to 1750. Clim. Dyn. 34, 1115–1128.

Noone, S., Broderick, C., Duffy, C., Matthews, T., Wilby, R.L. and Murphy, C.: A 250-year drought catalogue for the island of Ireland (1765–2015), Int. J. Climatol., 37 (Suppl.1), doi: 10.1002/joc.4999, 2017.

Pauling, A., Luterbacher, J., Casty, C., and Wanner, H., 2006: 500 years of gridded high-resolution precipitation reconstructions over Europe and the connection to largescale circulation, Clim. Dyn. 26, 387-405.

---

## Author Comment (AC3) · 10 Jan 2018

**Reviewer Two – Dr. Ashcroft**

This paper describes an extension of a recent rainfall dataset for the Island of Ireland, taking the existing 1850–2016 dataset back to 1711 using a recovered summary found at the Irish Meteorological Service Met Eireann. The study is one of those great data rescue stories that shows the value of this work. The data are well described, with limitations and caveats presented appropriately. I particularly liked the use of bootstrapping to get a range of estimates for the annual average rainfall used to construct the full series, and the visualisation of trends. It's a shame that the authors can't find more information about how the recovered summary was developed, but their use of independent and somewhat related sources to verify the extended record is well done. The information about other long-term rainfall data in and around Ireland in this paper will also surely be useful for future researchers. I recommend that the article be published subject to the consideration of the small suggestions below.

We thank Dr. Ashcroft for her supportive comments and thorough and thoughtful review. We share her frustrations about the lack of further information on the original source, but there is nothing more we can do about this, at least given current knowledge of the authors.

It would be good to include a map of Ireland, and perhaps the wider UK region, showing the locations mentioned in the text (including the independent sources) for those unfamiliar with Ireland.

This was also mentioned by Reviewer 1 and we will include a map of the British-Irish Isles with the location of the comparison stations and the locations of all areas mentioned in the paper on the map.

The study talks more about the rainfall variability in summer and winter than that of autumn and spring, despite the fact that the winter record is arguably the least reliable in the first few decades. Is there a reason for this? Are these key seasons for water security in Ireland, or seasons that show particular sensitivity to large-scale features?

Winter and summer are most interesting in term of extremes of flood and drought, while winter in particular is sensitive to large scale features. We also focus on them because they raise interesting questions about the reconstruction – in particular the early winter series and the wet mid-1700s in summer. The autumn and spring series show good reconstructions throughout the record and are less interesting in discussion terms than winter and summer.

Using bootstrapping to estimate the AAR (and uncertainty) seems like a good idea to me. However, a reader might think that the uncertainty shown in Figure 2 represents all of the associated uncertainties with the data, including quality issues of the very early observations. You discuss this towards the end of the paper, but it would be good to add a disclaimer about this in Section 2.1.2 (and in the caption to Figure 2) to clarify.

Good point, we can reword and flag as follows:

In section 2.1.2 we will clarify the sentence relating to the uncertainty bounds as follows:

Second, confidence bounds can be generated for the reconstructed series to convey the uncertainty in estimating the value of AAR.

In the caption of Figure 2 we will describe the bounds as follows:

The grey shading shows the uncertainty in the reconstruction from resampling of the baseline used to estimate AAR only.

Is there any reason why you used Spearman rank correlation over Pearson?

Simply that Spearman's makes no assumption about the distribution of the underlying data. We will clarify.

In section 2.2.1, you mention that the EWR series was used to calibrate the Jenkinson series. Can you elaborate on that? Did you do that, or Jenkinson et al?

This was done by Jenkinson et al (1979) and relates to one of the aspects that we have little information on how the calibration was executed. We will alter the text to ensure clarity that this was done by Jenkinson et al. as follows:

Jenkinson et al. used the EWR data to calibrate their series for Ireland, so there are obvious circularities.

You mention where you obtained the Hoofddorp dataset, but no other comparison series. Are they all from the associated publications? Perhaps specify this.

We thank the reviewer for this valuable insight. We will include the following statement in section 2.2 and update any download links for series used.

Unless otherwise stated all datasets were obtained from authors of the associated papers.

Section 2.3: did you do any quality control/outlier analysis on the data before homogenisation?

The IoI series from 1850 -2015 was previously homogenised by Noone et al. (2016). Strictly speaking we do not homogenise the pre-1850 series, we do a basic quality control to check for evidence of breaks in the mean and variance but these are only to identify points of interest rather than correcting the series. Much of our discussion on winter in particular relates to a likely inhomogeneity in the data pre-1790, hence our focus on comparison with other long term records. We also highlight that a key direction for future work is to undertake reconstruction of the early series using regression methods to further examine this component of the record. We are not sure that such a long record, which at a minimum contains changes in recording practice, gauge design and contributing sources, can be called homogenous in a strict sense. We can only establish confidence in parts of the record, which we attempt to do in the paper.

Why did you use SNHT and the Pettitt tests for homogeneity assessment? Presumably it's because they don't need neighbouring stations, but it would be worth adding a sentence explaining your choice of these methods over more recent approaches such as RHtestV4.

Yes, these are single series tests and hence their appeal in this case given the lack of possibilities for relative homogenisation. We will add some justification for our choice of these methods over others.

It's great that the original data source is provided with the paper, but will the final digitised dataset also be made available?

Yes, if the paper is accepted we will provide a link to the data for downloaded, this will be held on Met Eireann's website. The link to the data will be provided in the Data Availability note with the paper.

A hopeful question: have you looked into whether Jenkinson or their colleagues are still alive?

We have. The obituary of Prof. Arthur Jenkinson, which outlines his considerable achievements, was published in Weather in December 2006. http://onlinelibrary.wiley.com/doi/10.1256/wea.171.04/pdf

As an aside: this work, while suitable for Climate of the Past, would also have been suitable for data-focussed publications such as Geoscience Data Journal or Earth System Science Data. This would have enabled you to attach a DOI to the Jenkinson record (and your dataset) and make it more prominent, rather than including it as an attachment.

Both journals were considered as a possible outlet for publication. We see the merits of attaching a DOI but concluded that the paper and hence the dataset would reach a wider audience and influence discussion more here.

In section 3.3.1 you mention that the post 1950 period is the wettest for winter, but in the discussion talk about the impact that increasing availability of observations has on the trend. Do you think that this post-1950 wet signal might also be an indicator of increasing data coverage?

In short, no – the number of contributing stations is static from 1850 to present.

Table 4: Two decimal places in the correlations is probably enough

We will revise.

Table 5 and 6: One caption mentions that these descriptions are 'derived from', while the other says 'taken from' Rutty's diary. Were they compiled in different ways?

No, both were taken directly from the Rutty diary. We will fix the wording to be consistent in both captions.

Figures 4 to 9: You might want to try some different colour schemes for these spaghetti type plots that are colour-blind appropriate (try http://colorbrewer2.org/).

We will do our best here, but this is going to be a challenge with so many comparison series. We are in full agreement that figures should be produced with such considerations in mind. One of the challenges is that we maintain a consistent colour for each series across all plots and will need to investigate possibilities. Adding different line styles (dashed/dotted) will obscure the purpose of the plots which is to show how the IoI series (the thick black line in the named figures) compares with other series.

The IoI curve should ideally by on top of the other too, I think that would improve readability.

This was our intention. We will reproduce all plots accordingly.

Finally, it would be great if you could spell out the acronyms used in the legend, at least in the first caption.

Thanks for highlighting this issue, the acronyms are already provided in Table 3 however we can include them in the first caption or make reference to the table.

*Technical corrections*

Abstract, line 10: I'd add the word 'boreal' before 'spring, summer and autumn', for southern hemisphere readers.

We will include in revisions

Abstract, line 14: add the word 'volcanic' before eruption

We will include in revisions

Page 3, line 8: do you have a reference about the lost diaries of William and Sam Molyneux?

Yes, it is already included and the reference will be made clearer in our revision.

Page 4, line 11: I'd add the word 'precipitation' before 'record'.

We will include in revisions

Page 4, line 21: I'd add 'community standard' or something similar ahead of the mention of HOMER, to signify its standing in the homogenisation field

We will include in revisions

Page 11, line 17: I feel like a word is missing at the start of this sentence. Maybe 'To derive the lol_1711 series MEAN'?

We will include in revisions

Page 11, line 23: you talk about the median and mean of the series in the same sentence, is that accurate?

Yes, we used the median series from resampling to examine for changes in the mean.

Page 12, line 28: I'd remove the 'For' at the start of this sentence

We will include in revisions

Page 17, line 35: Maybe add 'around that time' to the end of this sentence, and include a reference to George J Symons' network if you have one

We will include in revisions

Page 19, line 2: remove 'multiple' or 'different'

We will include in revisions

Page 19, line 5: add 'previously' before 'available'

We will include in revisions

Page 19, line 26: 'remain', rather than 'remains'

We will include in revisions

Page 19, line 5: add 'previously' before 'available'

Page 19, line 26: 'remain', rather than 'remains'

---

## Author Response (AR1)

**A 305-year continuous monthly rainfall series for the island of Ireland (1711-2016)**

**Murphy et al.**

We would like to thank both reviewers for their valuable comments. Below we respond to each reviewer in turn, highlighting changes made to the manuscript. Our responses are in blue. The tracked changes version of our revised manuscript is appended below. We have also given the manuscript a thorough review and double checked tables and figures etc. We also upload a clean version of abstract and manuscript following review.

**Reviewer 1 – Prof. Luterbacher**

This is an important contribution in deriving a continuous very long precipitation series for an area that is strongly influenced by the Atlantic and various circulation modes. The authors do a very good job in compiling all the meta information, the way how they generate the new series and provide associated uncertainties. They also carefully compare with other series and link statistically to upstream Atlantic climate conditions. I find this is an important contribution and I support it to be published in Clim Past. Below are a couple of suggestions/comments I wish the authors can take into consideration for the revisions.

We thank Prof. Luterbacher for the time taken to provide such a constructive review and for the positive comments above.

Abstract: The abstract might be improved. In my view it is too detailed with specific information that do not reflect the major findings of the study. It might include a synthesis of the findings and what makes this series unique. Also implication of the results and potential for future applications could be stressed.

In our revisions we have reworked the abstract as requested.

I suggest to provide a spatial correlation (spearman) plot between precipitation of Ireland (the representative station or country average) and Europe, separated for each season and for at least the last 50 years. That would show how the target new precipitation series is statistically linked to remote areas of the UK and European mainland.

An excellent suggestion that we hope will increase the uptake of our series in wider European work. We have included an additional figure and associated text (beginning of the results section on correlations) around interpretation in the revised manuscript.

I would suggest to produce a Figure that shows the location of the precipitation series that are compared with the target Ireland series. The starting year of the stations could be separated with different colors.

This was a point raised by both reviewers and we have plotted the locations of comparison series across the UK. Reviewer 2 also asked for locations from Ireland mentioned in the text to be included so that readers not familiar with the geography of the island can get a feel for the data/locations. We have now included a map as Figure 1 in the revised paper. One challenge however is that given the number of locations now included and the multitude of start dates it is very difficult to include start year by colour. This information is given in the tables (Table 1 and Table 2). In the manuscript and we refer to this in the figure caption.

Section 3.2., table 4 and fig 3: Please could you also calculate the significance for the running correlation. One way would be through bootstrapping. For the interpretation of this figure I would then only describe and interpret the significant periods.

We have done this in our revised manuscript. The following sentences have been added to the methods section and the plot and caption has been updated.

"For selected long-term series moving 30-year correlations were assessed. The 95% confidence levels for moving correlations were identified using a Monte Carlo procedure for which correlations between the observed and a set of one thousand randomly generated time series were estimated (Kokfelt and Muscheler, 2013). Confidence levels were calculated as the $2.5^{th}$ and $97.5^{th}$ percentiles of the moving correlation values returned by simulated series. Following Pauling et al. (2006), each time series generated by the Monte Carlo procedure has the same statistical attributes (variance, mean and lag-one autocorrelation) as the observations (Gershunov et al., 2001)."

Comparison with independent circulation: The authors might use the independent gridded SLP (and derived NAO-I, first EOF of gridded SLP) reconstruction by Küttel et al. (2010). They cover the past 260 years and are fully independent.

Thank you for this recommendation. The London SLP series, the Paris London Index and the Westerly Index also provide independent insights into SLP and the NAO-I. Given the number of series we already incorporate we examined the Küttel et al. (2010) dataset to determine if it was consistent with the aforementioned series and the extent to which it could add value to the paper (given the series already included). We focus attention on the derived NAO-I (leading EOF of the gridded Küttel et al. SLP data).

Agreement between the leading EOF for each season and the Hurrel PC based NAO-I is very strong for winter and spring (r>0.9), but less so for summer and autumn. This is most likely because the domain used in the Küttel et al. SLP series extends further east (to 50 deg) and includes the Arabian heat low (which will be very prominent in summer/autumn). Comparison of the winter (DJF) leading EOF shows a strong correlation with other modes of westerly flow already analysed in the paper. The table below shows Spearman's rank correlation matrix from 1760 – 2000 for winter (DJF) IoI rainfall and indices of westerly flow from our manuscript.

|        | IoI 1711 | WI   | PL Index | L-NAO | Küttel et al. EOF |
|--------|----------|------|----------|-------|-------------------|
| IoI_1711 | 1.00   | 0.50 | 0.31     | 0.46  | 0.32              |
| WI     | 0.50     | 1.00 | 0.52     | 0.43  | 0.56              |
| PL Index | 0.31   | 0.52 | 1.00     | 0.52  | 0.67              |
| L-NAO  | 0.46     | 0.43 | 0.52     | 1.00  | 0.71              |
| Küttel et al. EOF | 0.32 | 0.56 | 0.67 | 0.71 | 1.00          |

[Figure]

**Figure R1 Comparison of standardised decadal means for winter L-NAO, PL index and Küttel EOF**

Evident is the strong correlation between Küttel et al. and the PL-index and L-NAO index. Comparison of decadal means (standardised to period 1900-1950, as in our manuscript) also shows the strong coherence between the leading winter EOF from Küttel et al. and the PL Index and L-NAO index. We therefore conclude that integrating the Küttel et al. data into the paper adds little value above the series already included. Given the number of series being plotted already we would therefore rather not include it to maintain legibility of the plots. In addition, the other series, including an independent, homogenised SLP series for London, together with three estimates of westerly flow and the NAO-I, extend further back into the 1700s.

We have added the following sentence to the final paragraph of the discussion;

"In addition to the above independent sources, the gridded SLP dataset derived by Küttel et al. (2010) is a potential
additional independent source which we did not employ here but could prove useful in other work."

Apart from the NAO, the East Atlantic/Western Russia pattern (Eurasia-2 pattern EU2 (Barnston and Livezey, 1987) is one
of the most important modes for western Eurasia and shows also significant correlation with precipitation over Ireland. For
instance, the EU2 index measures the pressure difference across central Europe and thus is important in describing the
variability of Eurasian climate, especially during wintertime. Therefore the authors might also consider the EU index that has
been reconstructed back to AD 1675 (Luterbacher et al. 1999) and calculate the correlation and running correlation.
This is an excellent suggestion. The EU2 index is indeed highly correlated with our 1711 series, especially in winter. This is
a valuable index to our study and we have integrated it into the paper in our revisions. Specifically this is now included as
one of our independent comparison series and integrated into the assessment of running correlations, the correlations in
Table 4 and each of the panel plots for the seasonal comparisons. Thanks for bringing this to our attention.

It would also be interesting to compare your new reconstruction with those of Pauling et al. (2006) and Casty et al. (2007).
The authors provide 0.5◦x0.5◦ resolved gridded precipitation back to the mid 17th century (Pauling et al. 2006) and 1766
(Casty et al. 2007) including Ireland. The period could be analysed where there is no overlap of predictors. That would be
another benchmark test for the various datasets.
Again, an excellent suggestion. We have used the 1766 (Casty et al. 2007) series in previous work exploring historical
droughts in Ireland and have found it particularly useful (Noone et al. 2017) However for this study, both in terms of length
of record and independence of the early series the seasonal reconstructions by Pauling et al. (2006) are likely to be most
useful. Stations in Ireland are used as predictors in this series. Prior to this the Pauling et al. data would provide a useful
additional comparison series, though there remain issues of circularity which are impossible to avoid. In particular the
Jenkinson data was calibrated to EWR by the original authors, with some of the UK series that comprise Pauling et al., also
likely to be used for this purpose. Nonetheless we now integrate the Pauling et al. data into our revised manuscript and as a
series in our plots comparing running decadal totals with other precipitation series (Fig 6- Fig 10).

In using the Pauling et al data we extracted data for the grid cell closest to each of the Island of Ireland Precipitation
Network (IIP) (Noone et al. 2016) stations from 1711 onwards. Seasonal regression adjustments were then derived (over the
period 1850-2015) for each of the 25 stations comprising the IIP and applied to the Pauling et al. data representing each
station. The Pauling et al. Island of Ireland series was then constructed as the arithmetic mean from each of the 25 stations –

ie. in the same way as the IoI_1850 composite series (Noone et al. 2016). This detail is included in the revised manuscript when introducing the Pauling et al. data.

Different studies have shown, that the AMO/AMV are significantly correlated with precipitation/temperature downstream over the UK and Europe. It would be interesting to see whether the positive correlation within the instrumental period is also valid during the reconstruction period. There is new annual AMO/AMV reconstruction available that could be used (Wang et al. 2017). The data can be downloaded from this link https://www1.ncdc.noaa.gov/pub/data/paleo/reconstructions/wang2017/wang2017amvamo.txt

Thanks for this suggestion. As part of our revisions we analysed Spearman's correlations (at varying lags) of the Wang et al. (2017) reconstruction with IoI_1711 rainfall. There is a weak but significant positive correlation between the annually averaged data (r=0.2, >95% CI) at zero lag across the entire series (see Figure R2). The significant correlation at zero lag holds when the time series is split around 1860 (not far off 'instrumental period' definitions) (see Figure R2). Therefore, we find that the correlation pre and post our definition of instrumental is good and it does add further confidence to our series. This relationship and confidence holds on a seasonal basis (not shown). However, given the number of series in our revised manuscript already (which include an additional two following revision), and the relatively weak correlation (and hence explained variance) with AMO/AMV we opt not to include this analysis in the revised manuscript.

[Figure]

[Figure]

[Figure]

**Figure R2 Time series comparison of IoI_1711 and Wang et al. (2017) AMO/AMV reconstructions (left column) for the full series (1711-2010) (top), early record (1711-1859) (middle) and later record (1860-2010) (bottom). Lagged Spearman's correlations are also shown for the respective series (right column).**

5    It is striking, that the L-SLP is the only index that shows a significant negative correlation. Why is this?

In examining this comment we noticed that the running correlation plot (Fig 3 in our discussion paper) was wrong in our original submission. It repeated each plot for the same series and some other small issues. Corrections have now been applied to the plot and we also integrate the EU2 series given the previous excellent suggestion.

10   In relation to the comment, significant negative correlations are evident between CET and IoI/EWR rainfall in summer and with L-SLP and EU2 index in both seasons. In relation to SLP, this is due to blocking – high SLP over London is associated

with dry conditions over the islands. This is as expected. We would also expect the other indices to be positively correlated. We have added this clarity to the revised paper in section 3.2.

Decadal interpretations I suggest you also consider the Casty et al. (2007) and Küttel et al. (2011) papers with respect to linkages between the large scale atmospheric circulation and precipitation in Europe back to the mid-18th century.
We have integrated these useful additions into our discussion.

It would help the reader if the seasonal aspects and the comparison with other series could be shortly summarized, that would help the reader keeping the major features from the many numbers. This could be also in the form of a table that synthesis the results.
We have added and additional table (Table 5) to our revised manuscript for this purpose.

*Minor comments*

Page 6, top: Jenkinson et al. (1979) applied a graded scaling system, similar to Brázdil et al. (2010), to both diaries. In this context, please also cite Gimmi et al. (2007)
Done

Please note that Luterbacher et al. 2002 was published in 2001 and the publication date should be changed accordingly.
Done

All records were standardised (by mean and standard deviation) to the period 1900-1950 for visual comparison. Could you please specify why this period and if another more recent period would be chosen, if the results would be stable? Why only visually? What is the intention?
We choose this period given the availability of data across all series, together with the good quality of observations from the period. The results do remain valid for different periods for standardisation. We have removed the reference to 'visually', this was included given we are plotting multiple series on the same axis. The sentence has been edited to read:
"All records were standardised (rescaled to have a mean of zero and a standard deviation of one) to the period 1900-1950 to assess decadal variability and change. This period is common to all datasets and is a time of good data coverage and quality."

Concerning the precipitation conditions in 1816 in Ireland, please have a look at Veale and Endfield (2016) with additional information
We have integrated in the discussion paragraph that considers summer, as follows:

"In passing, we also note that summer 1816, the infamous 'year without a summer' following the Tambora eruption of 1815 (Luterbacher and Pfister, 2015), does not rank as notably wet in the IoI_1711 series (rank 53$^{rd}$ wettest). In line with Veale and Endfield (2016) we find that summer 1817 was wetter, ranking 4$^{th}$ wettest in our series."

5  Concerning the post precipitation conditions in Ireland after the Laki eruption, please have also a look at Brazdil et al. 2010
Thanks, we have added a citation to this paper in the discussion. Despite winter 1783-84 being exceptionally dry it is also associated with significant flood events, particularly in early January in Ireland, where there is evidence of rain on snow caused flooding. Following up this event there is ample evidence of extreme weather conditions across the island from floods to extreme cold and river freezing that we will follow up in another piece of research.

Potentially the following paper is of relevance and could be included/cited Evaluating the Dendrochronological Potential of Taxus Baccata (Yew) in southwest Ireland (Galvin et al. 2014)
We have cited.

15  The authors use Maunder Minimum as a key word but only mention it once. I would thus remove it.
Done

Additional references in email follow up post discussion phase with Prof. Luterbacher
Barriopedro, D., D. Gallego, M. C. Alvarez-Castro, R. García-Herrera, D. Wheeler, C. Peña-Ortiz, and S. M. Barbosa, 2014:
20  Witnessing North Atlantic westerlies variability from ships' logbooks (1685–2008). Clim. Dyn. 43, 939-955.
Wheeler, D., Garcıa-Herrera, R., Wilkinson, C.W. and Ward, C., 2010: Atmospheric circulation and storminess derived from Royal Navy logbooks: 1685 to 1750, Clim. Change, 101, 257-280, doi: 10.1007/s10584-009-9732-x
Moreno-Chamarro, E., Zanchettin, D., Lohmann, K., Luterbacher, J., and Jungclaus, J.H., 2017: Winter amplification of the European Little Ice Age cooling by the subpolar gyre. *Nature Sci. Reports*, **7**, 9981, DOI:10.1038/s41598-017-07969-0
25  We have included these references.

**Reviewer Two – Dr. Ashcroft**

This paper describes an extension of a recent rainfall dataset for the Island of Ireland, taking the existing 1850–2016 dataset back to 1711 using a recovered summary found at the Irish Meteorological Service Met Eireann. The study is one of those great data rescue stories that shows the value of this work. The data are well described, with limitations and caveats presented appropriately. I particularly liked the use of bootstrapping to get a range of estimates for the annual average rainfall used to construct the full series, and the visualisation of trends. It's a shame that the authors can't find more information about how the recovered summary was developed, but their use of independent and somewhat related sources to verify the extended record is well done. The information about other long-term rainfall data in and around Ireland in this paper will also surely be useful for future researchers. I recommend that the article be published subject to the consideration of the small suggestions below.

We thank Dr. Ashcroft for her supportive comments and thorough and thoughtful review. We share her frustrations about the lack of further information on the original source, but there is nothing more we can do about this, at least given current knowledge of the authors. We include a sentence in the last paragraph of the discussion that points to the possible valuable contribution that ongoing work to digitise and make available historical meteorological publication from the Met Office may have to this end.

"Finally, work is ongoing by Met Éireann and the Met Office National Library and Archive to digitally scan and improve the accessibility to a wide range of historical meteorological publications. For UKMO these are available through an open archive at https://digital.nmla.metoffice.gov.uk/. This resource may prove of considerable value to future climatological studies, not least in helping fill some of the gaps in our understanding of the original data and methodology for the Jenkinson Ireland series."

It would be good to include a map of Ireland, and perhaps the wider UK region, showing the locations mentioned in the text (including the independent sources) for those unfamiliar with Ireland.

This was a point raised by both reviewers and we have plotted the locations of comparison series across the UK. Reviewer 2 also asked for locations from Ireland mentioned in the text to be included so that readers not familiar with the geography of the island can get a feel for the data/locations. We have now included a map as Figure 1 in the revised paper. One challenge however is that given the number of locations now included and the multitude of start dates it is very difficult to include start year by colour. This information is given in the tables (Table 1 and Table 2) in the manuscript and we refer to this in the figure caption.

The study talks more about the rainfall variability in summer and winter than that of autumn and spring, despite the fact that the winter record is arguably the least reliable in the first few decades. Is there a reason for this? Are these key seasons for water security in Ireland, or seasons that show particular sensitivity to large-scale features?

Winter and summer are most interesting in term of extremes of flood and drought, while winter in particular is sensitive to large scale features. We also focus on them because they raise interesting questions about the reconstruction – in particular the early winter series and the wet mid-1700s in summer. The autumn and spring series show good reconstructions throughout the record and are less interesting in discussion terms than winter and summer.

Using bootstrapping to estimate the AAR (and uncertainty) seems like a good idea to me. However, a reader might think that the uncertainty shown in Figure 2 represents all of the associated uncertainties with the data, including quality issues of the very early observations. You discuss this towards the end of the paper, but it would be good to add a disclaimer about this in Section 2.1.2 (and in the caption to Figure 2) to clarify.

10 Good point, we have reworded and flag as follows:  In section 2.1.2 we clarify the sentence relating to the uncertainty bounds as follows:

"Second, confidence bounds can be generated for the reconstructed series to convey the uncertainty in estimating the value of AAR."

In the caption of Figure 2 (now Fig 3) we describe the bounds as follows:

15 "The grey shading shows the uncertainty in the reconstruction from resampling of the baseline used to estimate AAR only."

Is there any reason why you used Spearman rank correlation over Pearson?

Simply that Spearman's makes no assumption about the distribution of the underlying data. We emphasise this in the revised paper.

In section 2.2.1, you mention that the EWR series was used to calibrate the Jenkinson series. Can you elaborate on that? Did you do that, or Jenkinson et al?

This was done by Jenkinson et al (1979) and relates to one of the aspects that we have little information on how the calibration was executed. We will alter the text to ensure clarity that this was done by Jenkinson et al. as follows:

25 "Jenkinson et al. used the EWR data to calibrate their series for Ireland, so there are obvious circularities."

You mention where you obtained the Hoofddorp dataset, but no other comparison series. Are they all from the associated publications? Perhaps specify this.

Good spot. We include download links for series used. Where data were not available online we give the source.

Section 2.3: did you do any quality control/outlier analysis on the data before homogenisation?

The IoI series from 1850 -2015 was previously homogenised by Noone et al. (2016). Strictly speaking we do not homogenise the pre-1850 series, we do a basic quality control to check for evidence of breaks in the mean and variance but these are only to identify points of interest rather than correcting the series. Much of our discussion on winter in particular

relates to a likely inhomogeniety in the data pre-1790, hence our focus on comparison with other long term records. We also highlight that a key direction for future work is to undertake reconstruction of the early series using regression methods to further examine this component of the record. We are not sure that such a long record, which at a minimum contains changes in recording practice, gauge design and contributing sources, can be called homogenous in a strict sense. We can only

5   establish confidence in parts of the record, which we attempt to do in the paper.

Why did you use SNHT and the Pettitt tests for homogeneity assessment? Presumably it's because they don't need neighbouring stations, but it would be worth adding a sentence explaining your choice of these methods over more recent approaches such as RHtestV4.

10   Yes, these are single series tests and hence their appeal in this case given the lack of possibilities for relative homogenisation. We have added some justification for our choice of these methods Vs relative homogenisation methods.
Out of interest, as a follow up to the reviewers comment we also undertook testing using RHtestV3. No breaks in the mean were found for Spring, Summer and Autumn. In winter, breaks associated with the wet decade of the 1730s (see Figure R3) were found but our paper deals comprehensively with adding confidence to this period. Interestingly the RHtestV3 method

15   did not reveal any other breaks in the winter series, even if we start the test in 1740.

[Figure]

**Figure R3 Results of the application of RHtestV3 which identifies statistically significant breaks in the winter series in 1722 and 1739.**

It's great that the original data source is provided with the paper, but will the final digitised dataset also be made available?

Yes, we provide a link to Met Éireann's website in the newly added Data Availability section where the IoI_1711 monthly series can be downloaded.

5   A hopeful question: have you looked into whether Jenkinson or their colleagues are still alive?

We have. The obituary of Prof. Arthur Jenkinson, which outlines his considerable achievements, was published in Weather in December 2006. http://onlinelibrary.wiley.com/doi/10.1256/wea.171.04/pdf. We now add an additional acknowledgement to Prof. Jenkinson and colleagues for the work they did on the original series and include a reference to the obituary.

10  As an aside: this work, while suitable for Climate of the Past, would also have been suitable for data-focussed publications such as Geoscience Data Journal or Earth System Science Data. This would have enabled you to attach a DOI to the Jenkinson record (and your dataset) and make it more prominent, rather than including it as an attachment.

Both journals were considered as a possible outlet for publication. The Lead author sees the merits of attaching a DOI but concluded that the paper and hence the dataset would reach a wider audience and influence discussion more here.

In section 3.3.1 you mention that the post 1950 period is the wettest for winter, but in the discussion talk about the impact that increasing availability of observations has on the trend. Do you think that this post-1950 wet signal might also be an indicator of increasing data coverage?

In short, no – the number of contributing stations is static from 1850 to present.

Table 4: Two decimal places in the correlations is probably enough

Revised to two decimal places

Table 5 and 6: One caption mentions that these descriptions are 'derived from', while the other says 'taken from' Rutty's

25  diary. Were they compiled in different ways?

No, both were taken directly from the Rutty diary. We have fixed the wording to be consistent in both captions.

Figures 4 to 9: You might want to try some different colour schemes for these spaghetti type plots that are colour-blind appropriate (try http://colorbrewer2.org/).

30  We have redrawn all our 'spaghetti' plots using colours that are sensitive to colour-blindness. Our decisions were informed by this discussion http://jfly.iam.u-tokyo.ac.jp/color/#see, with our selected colours coming from Fig 16. We now use a palette of eight colours with three variants of line types.

The IoI curve should ideally by on top of the other too, I think that would improve readability.

This was our intention; we have revised all plots to ensure this is now the case.

Finally, it would be great if you could spell out the acronyms used in the legend, at least in the first caption.

This is tricky in the caption as each season has different series selected for plotting. Instead, in each caption we direct the reader to Table 3 for the explanation of acronyms used. We also do so for Table 4.

*Technical corrections*

Abstract, line 10: I'd add the word 'boreal' before 'spring, summer and autumn', for southern hemisphere readers.

Done

Abstract, line 14: add the word 'volcanic' before eruption

Done

Page 3, line 8: do you have a reference about the lost diaries of William and Sam Molyneux?

Yes, this was the Shields (1983) reference which now explicitly relates to this statement.

Page 4, line 11: I'd add the word 'precipitation' before 'record'.

Done

Page 4, line 21: I'd add 'community standard' or something similar ahead of the mention of HOMER, to signify its standing in the homogenisation field

Done

Page 11, line 17: I feel like a word is missing at the start of this sentence. Maybe 'To derive the IoI_1711 series MEAN'?

To include the word mean here is not correct. We leave as it is in this instance.

Page 11, line 23: you talk about the median and mean of the series in the same sentence, is that accurate?

Yes, we used the median series from resampling to examine for changes in the mean.

Page 12, line 28: I'd remove the 'For' at the start of this sentence

Disagree here, removal of 'For' makes the sentence read 'Comparison correlations of each variable..' which may give the sense that comparison correlations are some different form of statistic.

Page 17, line 35: Maybe add 'around that time' to the end of this sentence, and include a reference to George J Symons' network if you have one

Have altered the sentence, we would argue that our reference to the British Rainfall series encapsulates the work of Symons.

Page 19, line 2: remove 'multiple' or 'different'

Done

Page 19, line 5: add 'previously' before 'available'

Done

Page 19, line 26: 'remain', rather than 'remains'

Done

**Response to short comment from C. Mock**

Very nice and interesting paper, with a very important time series. I think the Royal Navy had a big presence at Cork starting around the late 1700s, and perhaps later work could incorporate the original ship logs from there.

5   Many thanks for taking the time to read and comment on our paper. You are right. Tyrrell (1995) did some interesting work with navy log books, but we are sure there is more that could be done here. We make note of this potential avenue for future work in our revisions.

References cited in author responses

10  Casty, C., Raible, C.C., Stocker, T.F., Wanner, H., Luterbacher, J.: European climate pattern variability since 1766, Clim. Dyn., 29, 791-805, 2007.
Gershunov, A., Schneider, N., Barnett, T.: Low-frequency modulation of the ENSO–Indian monsoon rainfall relationship: Signal or noise? J. Clim. 14, 2486–2492, 2001.
Kokfelt, U., Muscheler, R.: Solar forcing of climate during the last millennium recorded in lake sediments from northern
15  Sweden. Holocene, 23, 447–452, 2013.
Küttel, M., Xoplaki, E., Gallego, D., Luterbacher, J., Garcia-Herrera R., Allan, R., Barriendos, M., Jones, P.D., Wheeler, D., Wanner, H.: The importance of ship log data: reconstructing North Atlantic, European and Mediterranean sea level pressure fields back to 1750. Clim. Dyn. 34, 1115–1128, 2010.
Lawson, S.: Arthur Frederick Jenkinson. Weather, 60, 27. doi:10.1256/wea.171.04, 2005.
20  Luterbacher, J. and Pfister, C.: The year without a summer. Nat. Geosci., 8, 246-248, doi:10.1038/ngeo2404, 2015.
Noone, S., Broderick, C., Duffy, C., Matthews, T., Wilby, R.L., Murphy, C.: A 250-year drought catalogue for the island of Ireland (1765–2015), Int. J. Climatol., 37 (Suppl.1), doi: 10.1002/joc.4999, 2017.
Noone, S., Murphy, C., Coll, J., Matthews, T., Mullan, D., Wilby, R.L. and Walsh, S.: Homogenization and analysis of an expanded long-term monthly rainfall network for the Island of Ireland (1850–2010), Int. J. Climatol., 36(8), 2837-2853, doi:
25  10.1002/joc.4522, 2016.
Pauling, A., Luterbacher, J., Casty, C., and Wanner, H.: 500 years of gridded high-resolution precipitation reconstructions over Europe and the connection to largescale circulation, Clim. Dyn. 26, 387-405, 2006.
Tyrrell, J. G.: Paraclimatic statistics and the study of climate change: The case of the Cork region in the 1750s, Clim. Change, 29, 231–245, 1995.
30  Veale, L and Endfield, G.H.: Situating 1816, the 'year without summer', in the UK. Geogr. J., 182 (4), 318 – 330, doi: 10.1111/geoj.12191, 2016.
Wang, J., Yang, B., Ljungqvist, F.C., Luterbacher, J., Osborn, T.J., Briffa, K.R. and Zorita, E. Internal and external forcing of multidecadal Atlantic climate variability over the past 1,200 years. Nature Geoscience 10, 512-517, 2017.

**A 305-year continuous monthly rainfall series for the Island of Ireland (1711-2016)**

Conor Murphy[1], Ciaran Broderick[1], Timothy P. Burt[2], Mary Curley[3], Catriona Duffy[1], Julia Hall[4], Shaun Harrigan[5], Tom K.R. Matthews[6], Neil Macdonald[7], Gerard McCarthy[1], Mark P. McCarthy[8], Donal Mullan[9], Simon Noone[1], Timothy J. Osborn[10], Ciara Ryan[1], John Sweeney[1], Peter W. Thorne[1], Seamus Walsh[3], Robert L.Wilby[11]

[1] Irish Climate Analysis and Research UnitS (ICARUS), Department of Geography, Maynooth University, Ireland.

[2] Department of Geography, Durham University, Durham, DH1 3LE, UK and Department of Geographical Sciences, University of Bristol, Bristol, BS8 2LR, UK.

[3] Climatology and Observations Division, Met Éireann, Dublin, Ireland.

[4] Institute of Hydraulic Engineering and Water Resources Management, Technische Universität Wien, Vienna, Austria.

[5] Centre for Ecology & Hydrology, Wallingford, Oxfordshire, OX10 8BB, UK.

[6] School of Natural Sciences and Psychology, Liverpool John Moores University, Liverpool, Merseyside, L3 3AF, UK.

[7] Department of Geography and Planning, School of Environmental Sciences, University of Liverpool, Liverpool, UK.

[8] Met Office, Hadley Centre, Fitzroy Road, Exeter, EX1 3PB, UK.

[9] School of Natural and Built Environment, Queen's University Belfast, N. Ireland, UK.

[10] Climate Research Unit, School of Environmental Sciences, University of East Anglia, Norwich, UK.

[11] Department of Geography, Loughborough University, UK.

*Correspondence to*: Conor Murphy (conor.murphy@mu.ie)

**Abstract**

A continuous 305-year (1711-2016) monthly rainfall series (IoI_1711) is created for the Island of Ireland. The post 1850 series draws on an existing quality assured rainfall network for Ireland, while pre-1850 values come from instrumental and documentary series compiled, but not published by the UK Met Office. ~~Two overlapping data sources are employed: i) a previously unpublished UK Meteorological Office note containing annual rainfall anomalies and corresponding proportional monthly totals based on weather diaries and early observational records for the period 1711-1977 and; ii) a long-term, homogenised monthly rainfall series for the island of Ireland for the period 1850-2016. Using estimates of long-term average precipitation from the homogenised series to merge these sources, the new 305-year record is constructed and insights drawn about notable extremes, climate variability and change.consistency of the resultingwasthe record amongst all series in, summer andIn the annual seriesfromto therecordsnew record Island of Ireland series revealsallthestrongThe 
[revised manuscript text omitted]